# Solar cycle as a distinct line of evidence constraining Earth's transient climate response

King-Fai Li [1] & Ka-Kit Tung [2] ✉

Severity of warming predicted by climate models depends on their Transient Climate Response (TCR). Inter-model spread of TCR has persisted at ~100% of its mean for decades. Existing observational constraints of TCR are based on observed historical warming response to historical forcing and their uncertainty spread is just as wide, mainly due to forcing uncertainty, and especially that of aerosols. Contrary, no aerosols are involved in solar-cycle forcing, providing an independent, tighter, constraint. Here, we define a climate sensitivity metric: time-dependent response regressed against time-dependent forcing, allowing phenomena with dissimilar time variations, such as the solar cycle with 11-year cyclic forcing, to be used to constrain TCR, which has a linear time-dependent forcing. We find a theoretical linear relationship between the two. The latest coupled atmosphere-ocean climate models obey the same linear relationship statistically. The proposed observational constraint on TCR is about 1/3 as narrow as existing constraints. The central estimate, 2.2 °C, is at the midpoint of the spread of the latest generation of climate models, which are more sensitive than those of the previous generations.

Comprehensive climate models, which incorporate more and more known relevant physics, have become more sensitive, as measured by their warming response to doubling atmospheric $CO_2$ experiments in the latest generation, 6th Coupled Model Intercomparison Project (CMIP6)[1]. Tighter observational constraint is needed to determine the plausibility of the more sensitive models[2]. The climate science community has devoted over four decades to constrain a canonical climate sensitivity metric, the Equilibrium Climate Sensitivity (ECS), which is the global surface warming at equilibrium to doubling atmospheric $CO_2$ concentration. At equilibrium there should not be any ocean heat uptake; extrapolating from our current state with large but inadequately measured ocean uptake to that equilibrium state involves uncertainties. Another climate sensitivity metric, the Transient Climate Response (TCR), is defined as the global surface temperature response to 1% per year increase in atmospheric $CO_2$ at the time of doubling. The idealized forcing is to standardize the forcing in the model runs for this experiment. The model produced warming in response to it

depends on each model's climate sensitivity: the more sensitive the model is, the higher TCR it produces. TCR is more relevant for projections and mitigation decisions within a century, and more closely related to the social cost of carbon[3], compared to ECS. It has been estimated that halving the range of uncertainty for TCR has a present economic value of $10 trillion[4].

Combining multiple lines of evidence can often achieve a reduction of uncertainty for the estimates of climate sensitivity and that strategy has been successfully applied to ECS[5] by combining three lines of evidence, with historical warming being one of them, and the other two being from paleo climates. There is however a lack of truly independent lines of evidence for TCR other than historical warming, because, unlike ECS, TCR involves a specific time-dependent forcing, and there are no other known analogs. Here, by proposing a regressed climate sensitivity metric, dissimilar time-variations of forcing can be incorporated, and this allows more independent phenomena to be included as lines of evidence. We demonstrate here its use in

[1]Department of Environmental Sciences, University of California, Riverside, CA, USA. [2]Department of Applied Mathematics, University of Washington, Seattle, CA, USA. ✉e-mail: ktung@uw.edu

constraining the TCR utilizing the observed response to solar-cycle forcing.

The existing line of evidence uses the observed historical global warming[6–8] to directly constrain the TCR of model simulated warming. The dominant uncertainty in this approach lies in the different forcings used in different models to reproduce the observed historical warming. Largely because of the uncertain aerosol forcing in models, model simulated warming is not a good discriminator of model sensitivities, because most climate models succeeded in simulating the same (known) centennial warming, despite their widely different climate sensitivities (albeit more so for ECS than for TCR)[9]. Recent improvements in estimating the constraint involve turning the historical warming in models into an "Emergent Constraint" for TCR by seeking a linear relationship for shorter, post-1975 decades that had more constant aerosol loading, with empirically fitted slope and intercept[10,11]. On the other hand, climate models have not been tuned to simulate the solar-cycle forcing. As a result, solar responses are found to differ widely from model to model, with larger response generally for higher TCR. Consequently, this phenomenon may provide a more distinct discriminator for model TCRs, and—without the "shared bias" of attempting to simulate the same well-known observation—may be more suitable for use in "Emergent Constraints". Furthermore, solar cycle has a separate external forcing than $CO_2$. The response to this external forcing is amplified by the climate feedback processes of the Earth system. It is hoped that by examining the solar-cycle response at the Earth's surface, we can estimate the effect of the climate feedback involved, and hence the climate sensitivity of the Earth system. This is a different approach than some of the current ones, for example, of using the effect of Arctic snow-albedo[12], or tropical low clouds[13,14] to provide a constraint on the climate sensitivity (ECS in the cited cases). Using a component of the feedback process to constrain ECS or TCR requires an additional assumption that the other feedback components are "unbiased", e.g. not compensating[9]. This structural uncertainty has so far not been assessed.

Nevertheless, arguments against using the solar-cycle forcing are: (a) its cyclic-in-time forcing is very different than the linearly increasing forcing of the TCR experiment; (b) there is a difference in the lag of response to forcing introduced by the deeper ocean because of the difference in time scales in the two phenomena; (c) whether the two phenomena experience the same climate feedbacks, and (d) methods used to extract the expected small solar-cycle signal need to be particularly effective in reducing contamination by internal variability and other forcings. A modified climate sensitivity metric, which is the time-dependent response regressed against time-dependent forcing, is defined here to overcome problem (a). We will use both 2-layer emulators of climate models and the climate models themselves in discussing why we think difficulties (b) and (c) can be overcome. To address (d), we will describe and test a more sophisticated space-time method to extract the solar-cycle response.

## Results

### Solar-cycle response extracted from instrumental record

The 11-year solar cycle, also called the 11-year sunspot cycle, is a quasi-periodic phenomenon associated with dark spots on the surface of the Sun. The brightening effect in the surrounding faculae overcompensates the dimming effect of the dark spots, producing more radiation during the solar maxima (solar max) than solar minima (solar min). This variation between solar min and solar max in the Total Solar Irradiance (TSI) has been measured by orbiting satellites since late 1978 to be variable between cycles but close to 1 W m$^{-2}$, in a disk facing the sun at the top of the atmosphere. (TSI is defined as the spectrally integrated power from the sun received at the top of the Earth's atmosphere. TSI/4 is the power per unit spherical area of the Earth). The relative accuracy (the variation between solar max and solar min) is high, but because of changing satellites, the long-term variability in TSI is more uncertain and has been

a subject of debate[15]. Nevertheless, the secular trend is small[16], and we are concerned only with the relative variation. TSI before the satellite era has been reconstructed based on proxies of the solar dark and bright magnetic regions. The uncertainties, mostly involving the trend, are discussed in Methods, and found to be small.

Using the most recent observed surface temperature dataset HadCRUT5[17], which provided 200 ensemble members to assess various aspects of observational uncertainty, we calculate the global mean solar-cycle response $\kappa_{SOLAR}$ as the regression coefficient against TSI, denoted by $\langle T_{SOLAR}(t)|TSI(t)\rangle$. (See Methods for how regression is calculated analytically and numerically).

The mean value is obtained as the ensemble mean (over the 200 surface temperature sets) for the modern period 1956–2014 (see Fig. 1), involving 5 complete solar cycles, with brackets denoting the "very likely" range (5–95%):

$$\kappa_{SOLAR} = 0.084 \, [0.070 \text{ to } 0.097] \, ^{\circ}\text{C/W m}^{-2} \tag{1}$$

The two groups used in the Linear Discriminant Analysis (LDA) analysis are the solar max group and the solar min group. The years with TSI above (below) the local mean are classified into the solar max (min) group. The local mean is determined using Empirical Mode Decomposition[18,19] (see Methods). The latest climate models, CMIP6, ended their historical runs after 2014. Since we want to use the same period for both observation and model outputs, 2014 is the last year we can use. To avoid known systematic bias of our method from having uneven number of years in the solar max group and solar min group when we extract the solar signal using our LDA method, complete solar cycles need to be used (with at most one-year difference between the two groups). It is the nature of the solar-cycle phenomenon that there usually are more years in solar min group than in the

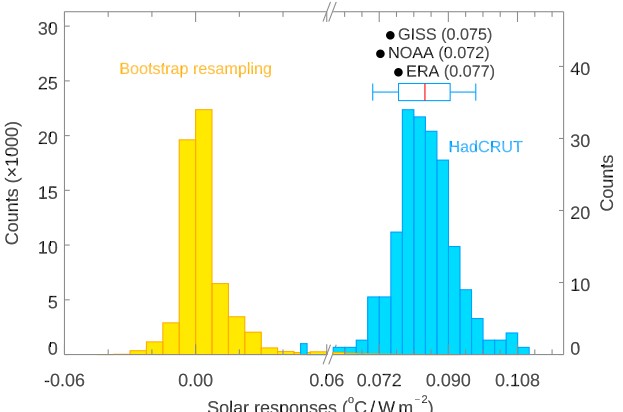

**Fig. 1 | Solar-cycle response, $\kappa_{SOLAR}$, extracted by the Linear Discriminant Analysis[61,63] (see Method).** The blue histogram, with expanded horizontal scale, shows the distribution of global solar response regressed against the Total Solar Irradiance, $\kappa_{SOLAR}$ of 200 HadCRUT5 ensemble runs[17]. The box and whisker summarize the 17–83% and 5–95% ranges of the HadCRUT5[17] distribution, respectively. The black dots are the values of $\kappa_{SOLAR}$ for three other datasets: GISS[44,71] and NOAA[72,73], which are geographically complete (by in-filling), and ERA[45,46], which is also geographically complete but is a reanalysis. ERA is used here to assess the difference between Sea-Surface Temperature (SST) and 2-m surface temperature (see Methods). All observation data cover 1956–2014 except ERA, which covers only 1960–2004 due to data availability. The yellow histogram shows the "null distribution" of $\kappa_{SOLAR}$ obtained by bootstrap resampling of the real data and then applying the same Linear Discriminant Analysis (LDA) method (see Methods and Supplemental Information) as that used to obtain the blue result. A total of 60,000 bootstrap resamples are drawn from the 200 HadCRUT5 ensemble members. An L-block resampling of 11 years is applied to take into account autocorrelation. Note the expanded left vertical scale for the yellow bootstrap samples.

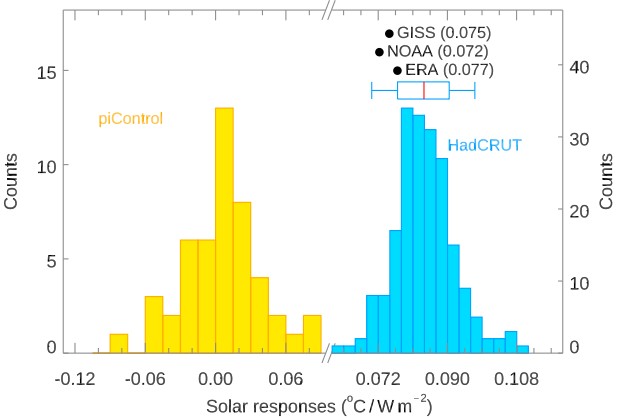

**Fig. 2 | Solar-cycle response vs PreIndustrial Control.** Same as Fig. 1, except the observed solar-cycle responses ($\kappa_{SOLAR}$) are compared against the Pre-Industrial Control (piControl) runs in CMIP6. The yellow histogram shows the "null distribution" of $\kappa_{SOLAR}$ simulated without solar-cycle forcing for all 48 CMIP6 models in the piControl runs. It is distributed around zero with both positive and negative values, similar to that from a random distribution. The larger negative values are from models that do not have adequate spin-up of their oceans. See Supplementary Table S1.

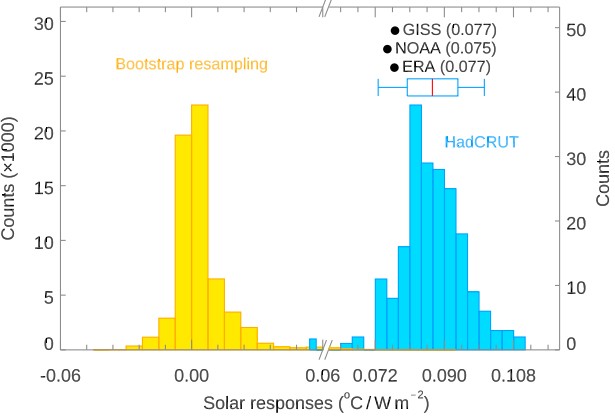

**Fig. 3 | Solar-cycle response obtained by excluding years of volcano eruption.** Same as Fig. 1 except the solar responses ($\kappa_{SOLAR}$) are calculated by excluding the two years after the Pinatubo and after El Chichón volcanic eruptions. The blue histogram, with expanded horizontal scale, shows the distribution of global solar response regressed against the Total Solar Irradiance, $\kappa_{SOLAR}$ of 200 HadCRUT5 ensemble runs[17]. The box and whisker summarize the 17–83% and 5–95% ranges of the HadCRUT5[17] distribution, respectively. The black dots are the values of $\kappa_{SOLAR}$ for three other datasets: GISS[44,71] and NOAA[72,73], which are geographically complete (by in-filling), and ERA[45,46], which is also geographically complete but is a reanalysis. All observation data cover 1956–2014 except ERA, which covers only 1960–2004 due to data availability. The yellow histogram shows the "null distribution" of $\kappa_{SOLAR}$ obtained by bootstrap resampling of the real data and then applying the same Linear Discriminant Analysis (LDA) method (see Methods and Supplemental Information) as that used to obtain the blue result. A total of 60,000 bootstrap resamples are drawn from the 200 HadCRUT5 ensemble members. An $L$-block resampling of 11 years is applied to take into account autocorrelation. Note the expanded left vertical scale for the yellow bootstrap samples.

solar max group, as solar max is usually reached rather quickly. Starting the period earlier than 1956 will increase the number of excess years in the solar min group, until 1946 (with 34 maxes and 35 mins). However, global observational data prior to 1950 are sparser. Therefore, we select 1956–2014 for the present study and the maximum number of solar cycles that we can use is five.

In addition, Fig. 1 shows the null distribution obtained using the method of bootstrap with replacement, by randomly scrambling the years of the observed data relative to the TSI[20]. Autocorrelations up to 11 years is taken into account by randomly sampling, with replacement, blocks of the time series each of length $L = 11$ years using the $L$-block method[21]. The extracted solar-cycle signal in Eq. (1) is statistically significant at over 99% confidence level. Furthermore, compared to the "null distribution" obtained from the solar-cycle "signals" in the 48 CMIP6 Pre-Industrial Control runs known to have no solar-cycle forcing, the observed signal is statistically significant at 100% confidence level. See Fig. 2. That is, no solar response is found by our method when it should not exist. Sensitivity to major volcano eruptions is discussed in "Methods" and Fig. 3 and found to be small.

## Solar cycle and TCR

As the only known phenomenon with a well-measured radiative forcing in the decadal range, the observed global response to the 11-year solar cycle can serve to constrain TCR for the following reasons:[22]

(1) The spatial patterns of the tropospheric and surface temperature response to 2% solar forcing and to $2\times CO_2$ are nearly the same in general circulation models[23], as originally proposed by Manabe and Wetheral[24] for persistent forcing. The observed latitudinal pattern of the 11-year solar cycle is also nearly the same, within observational uncertainty, as that predicted for $CO_2$ induced global warming and TCR[25]. As for the latest climate models, we compare in Fig. 4 the spatial structure of the TCR, calculated based on 41 CMIP6 models that have archived TCR runs in the open domain, with the spatial structure of the solar-cycle response calculated based on 51 CMIP6 models that have the solar cycle forcing in their historical runs. They are very similar, though with different amplitudes, consistent with Manabe and Wetheral[24]. The common features include: general warming over the globe, Arctic amplification of warming, more warming of the continents than over the oceans. And more warming over the

Arctic than over the Antarctic, the latter being more affected by the cold ocean upwelling than radiation from above.

(2) Effective Radiative Forcing (ERF) is defined as the effective radiative forcing at the top of the atmosphere after the stratosphere and troposphere has adjusted to the forcing while holding the surface temperature unchanged. A forcing agent with the same ERF as $CO_2$ should yield the same surface warming. ERF for the solar-cycle forcing has been calculated by the 6th Assessment Report (AR6)[9], putting it on the same footing as $CO_2$ when measuring their effects on surface warming, despite their very different long- and short-wave behaviors in the stratosphere. This will be discussed in more detail later.

(3) The fast climate radiative feedback mechanisms relevant for both phenomena appear to be the same, as deduced from their tropospheric patterns; the responses would both have been smaller without these climate feedbacks. We previously inferred from observations[26], and diagnosed explicitly using an aqua-planet model[23], that the radiative and dynamic feedback mechanisms for the solar-cycle response are: evaporative, water vapor plus lapse rate, cloud, and albedo feedbacks, same as those at play in the response to $CO_2$ forcing. The response to the solar radiative forcing contains these climate gains, which is measured by TCR[27]. This is likely the reason why when we normalize the smaller solar response by its smaller forcing, we find a linear correspondence with the TCR normalized by its larger forcing.

(4) Observational data analysis of the tropospheric solar-cycle response illustrates the pathway of the response from where the forcing is the largest to where the response is the largest[26]. Net radiative forcing at visible and infrared frequencies, which is the bulk of the solar-cycle forcing, is largest in the tropical region and yet the response at the surface is smallest there. The warming is largest over the polar region where the forcing is the smallest.

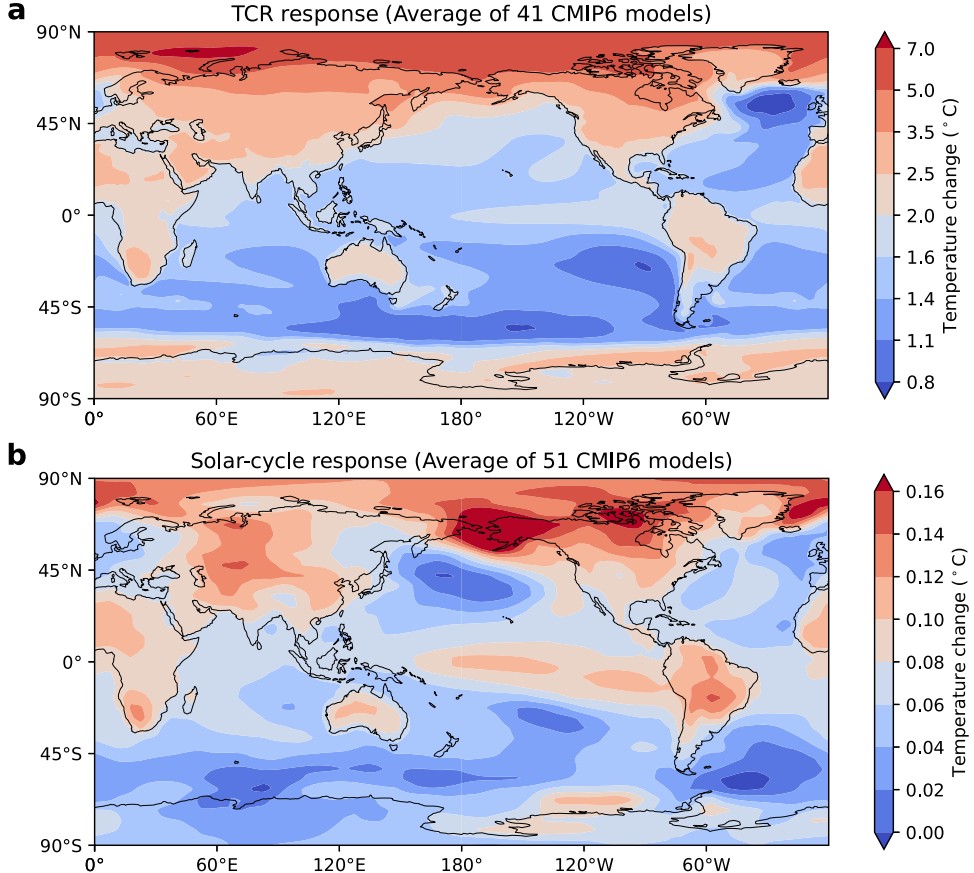

**Fig. 4 | Comparison of the surface temperature spatial structures. a** CMIP6 model Transient Climate Response (TCR) runs with 1% per year increase in $CO_2$. Shown is the mean spatial pattern of 60th–79th years minus the mean spatial pattern of 1st–10th years averaged over the 41 models with TCR data. **b** Solar cycle spatial pattern extracted by the Linear Discriminant Analysis (LDA) method as the pattern that best distinguishes the solar max group from the solar min group for the period 1956–2014 (see "Methods"). Shown is the average of 51 models with solar-cycle forcing. The LDA-projected times series $T_{SOLAR}(t)$ of each model is normalized to the unity regression coefficient $\langle T_{SOLAR}(t)|TSI(t)\rangle = 1$ so that the LDA

spatial pattern carries the unit of temperature. (The $T_{SOLAR}(t)$ used to obtain $\kappa$ in the text carries the unit of temperature while the LDA spatial pattern is normalized to a unity global mean). There may be a small contamination of the solar-cycle signal by the El Niño–Southern Ocean (ENSO) internal variability in the spatial pattern. This is removed in the second step of our spatial-time method for solar-cycle response extraction, whereby the time series of the response is regressed against the cyclic Total Solar Irradiance (TSI). This regression also removes remaining, if any, global warming contamination.

The tropical ocean surface is warmed little because of evaporative feedback. As water over the oceans is evaporated, heat is mostly carried aloft as latent heat instead of being absorbed by the ocean; the large ocean inertia does not come into play significantly to delay the response. We do not find more than a few months of lag. The latent heat of evaporation is deposited near 200–300 hPa in the tropical troposphere. There, down-gradient poleward heat transport aloft followed by thermal downwelling to the cold polar surface under the stable static stability of the polar lower troposphere combine to produce the characteristic spatial pattern of polar amplification of surface warming, the same mechanism as that for $CO_2$-induced warming. The mechanisms were analyzed[23] using an aqua-planet model and persistent forcing; it was found that the two phenomena share similar amplification mechanisms for the fast climate feedbacks in the troposphere although their stratospheric response is very different. We will show here that the relationship continues to hold for cyclic solar forcing.

## A different climate sensitivity metric

For time-varying forcing, we define a measure of sensitivity to external forcing as the regression of response to forcing, denoted as $\mu = \langle T(t)|F(t)\rangle$. The response is taken to be the global-mean surface temperature change, $T$, excluding unforced variability. The forcing, $F$,

is the global-mean ERF[9]. For solar-cycle forcing and response, we have $\mu_{SOLAR} = \langle T_{SOLAR}(t)|F_{SOLAR}(t)\rangle$.

$\mu_{SOLAR}$ is related to $\kappa_{SOLAR}$ through a scaling factor:

$$\mu_{SOLAR} = \frac{\partial TSI}{\partial F_{SOLAR}}\kappa_{SOLAR} \equiv b\kappa_{SOLAR} \qquad (2)$$

where from AR6[9], $\frac{1}{b} \equiv \langle F_{SOLAR}(t)|TSI(t)\rangle = \frac{0.72(1-\alpha)}{4}$, $\alpha$ being the planetary albedo. See later and Methods for assignment of uncertainties. Therefore, instead of finding $\mu_{SOLAR}$ using regression, we simply multiply $b$ to $\kappa_{SOLAR}$, which we have already obtained above, to get $\mu_{SOLAR}$.

## Normalized TCR

For the TCR experiment, a climate sensitivity metric can be defined as the global warming response regressed against its ERF:

$$\mu_{TCR} = \langle T_{TCR}(t)|F_{TCR}(t)\rangle$$

We additionally define the normalized TCR as:

$$\overline{TCR} \equiv \frac{TCR}{F_{2\times CO_2}} \qquad (3)$$

This quantity is not the same as $\mu_{TCR}$, but it is only slightly less (see later). When the regressed quantity cannot be calculated from the archived models, the normalized TCR is used.

Since the ERF from doubling $CO_2$, $F_{2 \times CO_2}$, is not an observed quantity, observations cannot constrain TCR itself, only the normalized TCR. AR6[9] assessed that the ERF for doubling $CO_2$ has a ±12% uncertainty. Using the normalized TCR avoids including this rather large uncertainty in our metric for TCR. We recommend models in the future report their TCR response regressed against the forcing, because the regression of the response to the linear forcing is better than simple ratios in reducing the contamination by other climate variability, in addition to absorbing the 12% uncertainty in the hypothetical forcing. We also recommend comparing the spreads in TCR by using the spread-to-mean ratio, where the numerator is the spread (5 to 95% range unless otherwise specified) and the denominator is the mean:

$$r_{TCR} = \frac{\Delta \overline{TCR}}{\overline{TCR}} \qquad (4)$$

Equation (4) can be applied directly to our normalized TCR, without having to first converting it to TCR.

Previously, direct observational constraints on TCR were provided by the "historical difference method", considering the ratio of historical change in global-mean temperature[6–8] since the preindustrial period, $\Delta T$, and in radiative forcing, $\Delta F$:

$$\mu_{TCR} \approx \mu_{HIST} = \frac{\Delta T}{\Delta F}$$

The uncertainty from the historical change method is large because (i) the temperature data in the beginning of the industrial era is sparse; (ii) the difference may include internal variability; and (iii) more importantly: "the single largest contribution to formal error in calculated TCR is, however, due to uncertainty in $\Delta F$"[8]. And the largest component of this uncertainty is due to aerosol forcing[7]. There is no aerosol involved in solar-cycle forcing, which we will use as a separate constraint.

## Expected linear relationship

We propose to use $\mu_{SOLAR}$ to constrain $\mu_{TCR}$. We expect a linear relationship between the two because both are amplified by the same climate gain factor on decadal time scales. This can be shown analytically in a 2-layer "emulator". Such emulators[9,28] were extensively used in AR6. It consists of a surface layer of temperature $T$, with heat capacity $C$, which is being forced by $F = Q(t)$, radiating to space above and losing heat to the deeper ocean layer below[28–30], whose temperature is denoted here as $T_d$ with heat capacity $C_d$:

$$C \frac{d}{dt} T = -\lambda T + Q - \gamma (T - T_d)$$
$$C_d \frac{d}{dt} T_d = \gamma (T - T_d) \qquad (5)$$

Numerical solutions are shown in Fig. 5. Exact analytic solutions are available for a linear forcing and for a periodic forcing, given by Geoffroy et al.[28], who also gave parameter values obtained by calibrating the 2-layer emulator with CMIP5 models. The solution consists of an instantaneous response to forcing plus slow and fast adjustments. Because of the disparity of time scales, the upper layer quickly adjusts, while the lower layer adjusts much slowly due to the larger

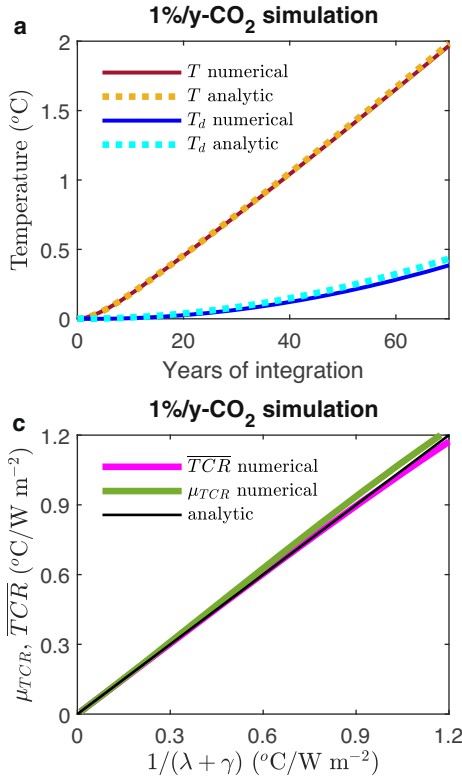

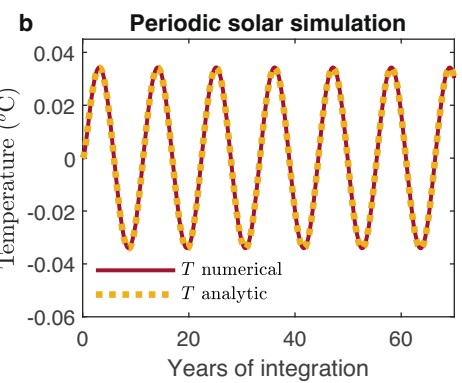

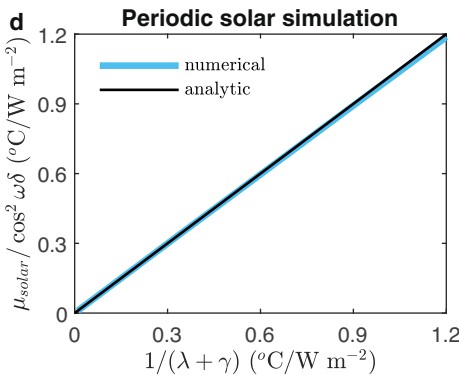

**Fig. 5 | Numerical and analytic solutions of the two-layer model.** Numerical and analytic asymptotic solutions of the full 2-layer model in Eq. (5). **a** Temperature response in the mixed layer and in the deep ocean of the solution with linearly increasing forcing. **b** Same as (a) but for the sinusoidal forcing solutions. **c** Linear relations in the numerical and analytic solutions with linearly increasing forcing.

**d** Same as (**c**) except for the sinusoidal forcing solutions. The approximate analytic solutions (Eq. (6ab)) are obtained using two-timing asymptotic method for the TCR forcing; for the solar-cycle forcing, it is obtained by ignoring deep ocean feedback onto the surface-layer temperature.

heat capacity of the deeper ocean[29]. We use an asymptotic solution, obtained using the two-timing method and verified to be accurate, to better show the dependence on these two timescales.

For the TCR experiment, the warming response to a switched-on radiative forcing

$Q_{TCR}(t) = q_{TCR} t$ for $t > 0$ and $Q_{TCR}(t) = 0$ for $t \leq 0$ is found to be:

$$T(t) = \frac{Q(t - \Delta)}{\lambda},$$

$$\text{where } \Delta \equiv \tau_0 \left(1 - e^{-t/\tau}\right), \tau_0 \equiv \frac{C_d}{\lambda}, \text{ and } \tau \equiv \left(\frac{\lambda + \gamma}{\gamma}\right) \tau_0, \quad (6a)$$

and

$$T_d(t) = \frac{Q(t - \Delta_d)}{\lambda} \quad \text{where } \Delta_d \equiv \tau(1 - e^{-t/\tau})$$

The instantaneous "equilibrium" response[28], $\frac{Q(t)}{\lambda}$, is the response obtained by ignoring ocean inertia. Due to ocean inertia, there is a lag of $\Delta(t)$ in the surface temperature response, starting at 0, reaching the asymptotic value of $\tau_0 \approx 80$ yr in $\tau \approx 240$ yr. The TCR experiment period is too short to reach that asymptotic value. The transient lag during the shorter TCR experiment period is linearly increasing, as $\Delta(t) \approx (\frac{\gamma}{\lambda + \gamma})t$, with the end value: $\Delta(70\,\text{yr}) \approx 20$ yr. The thermal parameters used in the estimates here are from the mean of the emulator model ensemble[28].

Regression of response to forcing yields:

$$\mu_{TCR} = \langle T_{TCR} | Q_{TCR} \rangle = \frac{\beta}{\lambda} \approx \frac{1}{\lambda + \gamma},$$

$$\text{where } \beta \equiv \frac{\partial(t - \Delta)}{\partial t} \approx 1 - \frac{\gamma}{\lambda + \gamma} = \frac{\lambda}{\lambda + \gamma} \quad (6b)$$

The normalized TCR can be found as:

$$\overline{TCR} = \frac{TCR}{q_{TCR} \times 70\,\text{yr}} = \frac{T(70\,\text{yr}) - T(0)}{q_{TCR} \times 70\,\text{yr}}$$

$$= \frac{Q(70\,\text{yr} - \Delta(70\,\text{yr}))}{\lambda \times q_{TCR} \times 70\,\text{yr}} \approx \frac{1}{\lambda + \gamma} \quad (6c)$$

The usual climate gain factor estimated using an energy-balance (one layer) model is $\frac{1}{\lambda}$, which is larger because ocean heat uptake is not taken into account.

Because the solar-cycle response does not penetrate through the whole depth of the mixed layer[31], it does not experience the full-depth heat capacity. We let $\eta$ be the fraction of the mixed layer depth that it penetrates to (to be determined from observation or constrained by the lag in the response). Since the signal does not penetrate the deeper ocean layer, $T_d$ is not affected by it, and can be set to be zero[30] in the approximate solution. For a sinusoidal solar-cycle forcing: $Q_{SOLAR} = q_{SOLAR} \sin(\omega t)$, the approximate solution is:

$$T_{SOLAR} = \frac{q_{SOLAR} \sin(\omega(t - \delta))}{(\lambda + \gamma)\sqrt{1 + \tan^2(\omega \delta)}} = \frac{q_{SOLAR} \sin(\omega(t - \delta)) \cos(\omega \delta)}{(\lambda + \gamma)},$$

$$\text{where } \tan(\omega \delta) \equiv \frac{\omega \eta C}{\lambda + \gamma}, \quad \text{and} \quad (7)$$

$$\mu_{SOLAR} = \langle T_{SOLAR} | Q_{SOLAR} \rangle = \frac{\cos(\omega \delta)}{\lambda + \gamma} \frac{\partial \sin(\omega(t - \delta))}{\partial \sin(\omega t)} = \frac{\cos^2(\omega \delta)}{\lambda + \gamma}$$

The analytical result in Eq. (7) is found to be very accurate compared to the full two-layer numerical solution (See Fig. 5). The lag is between 0 to 2 years for $\eta$ between 0 and 1. Especially accurate are the approximate expressions for the normalized TCR and also for the solar response regressed against forcing, which are being used in the expected linear relationship (see Fig. 5c, d). These relationships are independent of model parameters used in Fig. 5a, b.

Although the two phenomena are forced by separate forcings and the responses can be extracted separately, their sensitivity metrics are related by a linear relationship because they each are proportional to $\frac{1}{\lambda + \gamma}$; see Fig. 5c, d. Therefore:

$$\mu_{TCR} = \frac{1}{\lambda + \gamma} = a\,\mu_{SOLAR}, \quad \text{where } a = \frac{1}{\cos^2(\omega \delta)} \quad (8)$$

When the regressed form of the model climate sensitivity metric is not available, we use the normalized TCR:

$$\overline{TCR} = a\,\mu_{SOLAR} \quad (9)$$

The slope $a$ depends only on the lag of the solar response. It is close to 1 because the lag in the solar-cycle responses in observations is close to 0, subject to a maximum of 6-month uncertainty because annual-mean data are used.

## Linear relationship in CMIP6 models

54 CMIP6 models are available on esgf-node.llnl.gov. Meehl et al[1]. computed TCR for 37 models consistently using the ESMValTool. Zelinka et al.[32] computed the ERF in a set of 27 models; one (EC-Earth3) of these was not used in Meehl et al. Among the 26 common models, 6 (CAMS-CSM1-0, CESM2-WACCM, CNRM-CM6-1-HR, FGOALS-F3-L, INM-CM4-8, and SAM0-UNICON) did not follow CMIP6's protocol, which requires at least 500 years of spin-up of their oceans before TCRs are calculated (see Supplementary Table S1, and the discussion on the effect of not having sufficient spin-up on TCR in Methods, which can give rise to 30% uncertainty); they are objectively excluded. In addition, NORESM2-LM did not provide information on branching off time or preindustrial control, and so is also excluded. Information on MPI-ESM1-2-LR, which was not included in ref. 32, became available recently, and is added to our list. Sensitivity to including all 27 CMIP6 models is discussed in a later section and Fig. 10. CMIP5 models did not adopt a consistent spin-up protocol[33], thus introducing an additional but artificial decadal variability, and so are not used.

CMIP6 models seem to be consistent with Eq. (9). We use linear least-squares fit of the model pairs, and Monte-Carlo bootstrapping to find the spread of the slopes. The CMIP6 ensemble yields a regressed slope of 1.02 [0.97,1.16] for a best fit from the origin, consistent with Eq. (9) with slope 1 obtained above from a simpler model. This consistency test is not a proof; for that purpose, a CMIP6 model with many versions each with a different TCR is needed, which is however not available.

## Emergent constraint

Suppose we do not know the functional form $G(x)$ that relates the quantity $y$, to be constrained, to a quantity $x$, which could be observed: $y = G(x)$, but would like to discover it. The discovered relationship that "emerges" by exploring model pairs $(x, y)$ empirically is referred to as an "emergent constraint". Often because of the limited model points, the best one can do is through a local approximation around a typical point $(x_0, y_0)$ in a cluster of model data points: $y - y_0 \cong G'(x_0)(x - x_0)$, that is, a linear relationship of the form: $y = A x + B$. Model points are then regressed onto this line. The constant term $B = y_0 - G'(x_0)x_0$ is sometimes referred to as the "intercept". It is often nonzero in practice regardless of whether the true functional relationship passes through the origin. Both slope and "intercept" of the local approximation are affected by the scatter from the centroid of data points.

In Fig. 6 we empirically fit the model pairs using least-squares to this line. Bootstrapping with replacement generates 10,000 random sets of data with 20 points each. The 10,000 linear-least-squares fits form a probability density function, from which 5%–95% band is defined. The result obtained through robust regression remains the

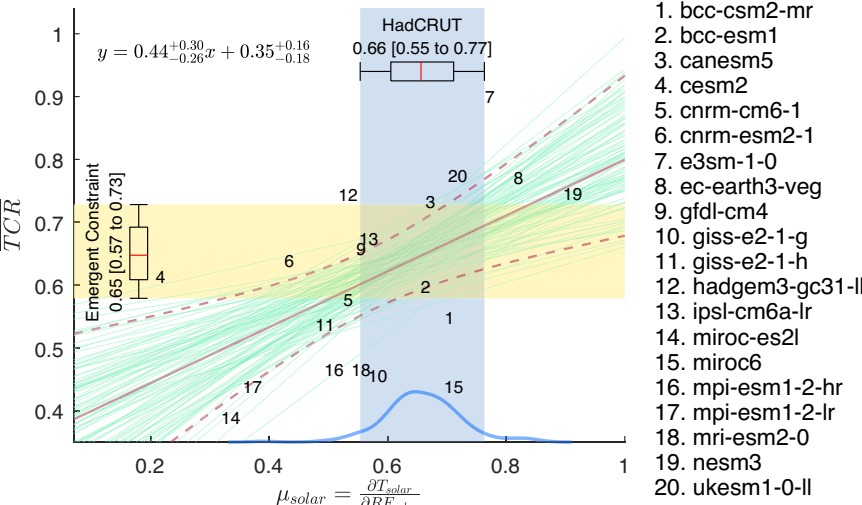

**Fig. 6 | Emergent Constraint.** CMIP6 model's normalized TCR (defined as $TCR/F_{2\times CO_2}$) vs 11-year global-mean solar-cycle response $\mu_{SOLAR}$ in 2-m temperature regressed against the Effective Radiative Forcing; both in units of °C/W m⁻². The shaded blue region indicates the spread of the observations. The regressed lines are drawn in green (only 50 are shown). The solid maroon line indicates the mean of the fit. The dashed maroon curves bound the 5–95% interval of the fit and same. Projecting the observed (blue band) onto this 5–95% range of slopes determined from model scatters yields should be interpreted as the prediction limits of the mean normalized TCR given an observed value of the solar response. The box and whisker represent the 17–83% and 5–95% "likely" and "very likely" intervals, respectively, obtained using 10,000 bootstrap samples of the 200 HadCRUT ensembles (each sample has 200 points with replacement), plus a Gaussian noise of $b$, which is the "error in variable". The super and subscripts indicate "very likely" interval of the estimated parameter.

same. Projecting the observed (blue band) onto this 5–95% range of slopes determined from model scatters yields

$$\overline{TCR} = 0.65\,[0.58\ \text{to}\ 0.73]\,°\text{C/W m}^{-2}\,;\ \ r_{TCR} = 23\% \qquad (10)$$

In the language of regression analysis, $x$ is the *predictor*, while $y$ is the *predictand*. The first value, 0.65, is the *predicted mean value* (or sometimes called the "best estimate"). The brackets enclose the 90% confidence level of that *predicted mean*; the maroon dashed lines in Fig. 6 indicate the *prediction limits of the mean*. This interval does not necessarily include 90% of the predicted model $y$ given model $x$. The latter is called[34] the 90% *predicted y interval* given $x$, and could be much wider. It is useful for a different purpose: to predict where *individual* model's TCR lies in 90% of the cases given a value of $x$. We are asking instead where the "best estimate" lies in 90% of the cases.

The centroid of the model data points happens to lie within the observed range (the blue vertical band). This is the narrowest part of 90% interval bounded by the dashed curves. The uncertainty range would have been larger if the observed range had occurred to the right or left of this narrow point. Also, the derived range is not sensitive to the slope of the regression lines since the various straight lines pivot near this point.

For comparison, recent emergent constraints on TCR using CMIP6 models based on simulated historical warming post-1975 gave 1.68 [1.0 to 2.3] °C by Nijsse et al.[11], and 1.60 [0.90 to 2.27] °C by Tokarska et al.[35]. AR6[9] combined these to claim 1.7 [1.1 to 2.3] °C. These have wider spreads of 77%, 86% and 71%, respectively. Ours, at 23%, is the narrowest, about 1/3 of existing constraints.

However, all three, including AR6's and ours, may be subject to a criticism of model-fitting bias, which is often raised in machine learning literature, but has so far not been considered in Emergent Constraint research. The two-parameter fit to the straight line used in the Emergent Constraints includes an "intercept", which is not physically justified but is employed solely for the purpose of yielding a better fit to the "training data". The "training data" here is the CMIP6 models. This may be a bias toward CMIP6 models.

Eliminating this bias, Emergent Constraint without an intercept should yield a wider range:

$$\overline{TCR} = 0.671\,[0.577\ \text{to}\ 0.833]\,°\text{C/W m}^{-2}\,;\ \ r_{TCR} = 38\% \qquad (11)$$

obtained for fits to lines through the origin (Fig. 7). The uncertainty range of our one-parameter fit is still ~1/3 of existing constraints: 120%[7] and 139%[8], when compared to other one-parameter fits of historical warming.

A still more conservative approach[34,36] makes no assumption about the existence of any relationship between $x$ and $y$, and suggests not leveraging models whose "observables" ($x$) are inconsistent with the current observed to "discover" a linear relationship. So only the model points within the blue band are fully weighted. Weighting of the points away from the blue band less using Kullback-Leibler divergence in information theory, a new probability density distribution is then constructed, from which it yields: $\overline{TCR} = 0.62\,[0.39\ \text{to}\ 0.86]\,°\text{C/W m}^{-2}\,;$ $r_{TCR} = 76\%$. See Fig. 8. This wider uncertainty range is not adopted here since we have established that a linear relationship does exist.

Another "line of evidence" uses direct observational constraint of warming in the instrumental record (without using the Emerging Constraint) to yield a $r_{TCR}$ of 120% by Lewis and Curry[7], and 139% by Richardson[8] (see "Methods"). Though their uncertainty ranges are much larger, these direct constraints were obtained without having to use an adjustable "intercept", and are therefore not subject to criticisms often leveled at some Emergent Constraints about not including "structural uncertainty"[37].

**Process-based estimate of TCR**

AR6[9] uses an energy-balance model for its process-based estimate. Here we use a two-layer model emulator, with its parameter calibrated against observations instead of models for the estimate. This approach does not use CMIP6 climate model pairs to empirically determine the relationship and thus avoids the possible criticism that we used some CMIP6 models which may "run too hot". We find almost the same result on the TCR constraint as above. It shows consistency with the Emergent Constraint approach but is probably not independent enough to be considered an additional line of evidence.

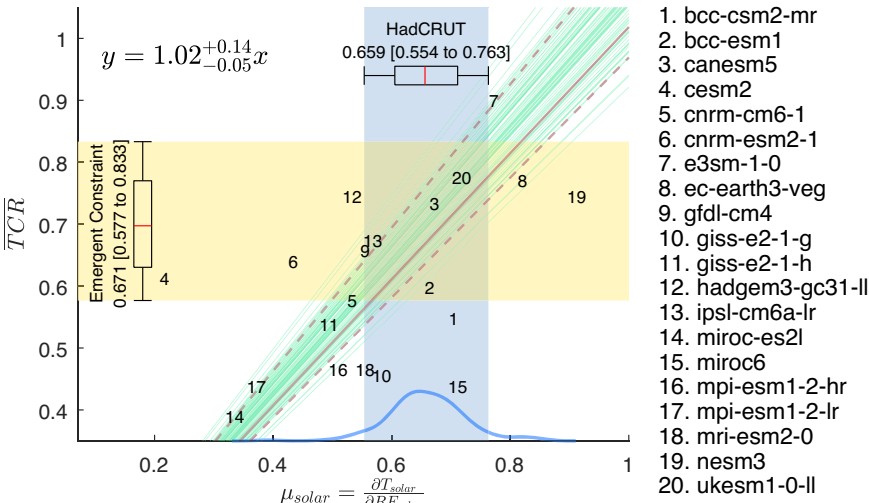

**Fig. 7 | Emergent Constraint without intercept.** Same as Fig. 6 except that the model points are fitted to a linear line without intercept. CMIP6 model's normalized TCR (defined as $TCR/F_{2\times CO_2}$) vs 11-year global-mean solar-cycle response $\mu_{SOLAR}$ in 2-m temperature regressed against the radiative forcing; both in units of °C/W m⁻². The shaded blue region indicates the spread of the observations. The regressed lines are drawn in green (only 50 are shown). The solid maroon line indicates the mean of the fit. The dashed maroon curves bound the 5–95% interval of the fit and

should be interpreted as the prediction limits of the mean normalized TCR given an observed value of the solar response. The box and whisker represent the 17–83% and 5–95% "likely" and "very likely" intervals, respectively, obtained using 10,000 bootstrap samples of the 200 HadCRUT ensembles (each sample has 200 points with replacement), plus a Gaussian noise of $b$, which is the "error in variable". The super and subscripts indicate "very likely" interval of the estimated parameter.

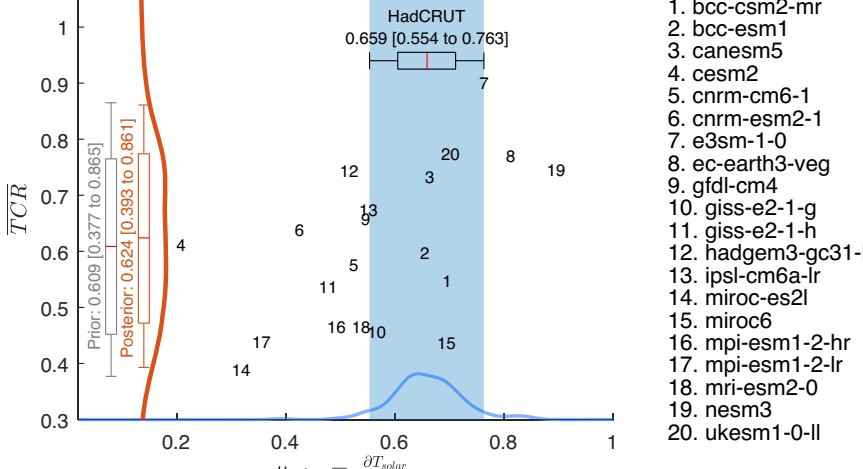

**Fig. 8 | Prior and posterior TCR ranges, following Brient and Schneider 2016 and Brient 2020.** The vertical box and whiskers are the 17–83% and 5–95% ranges of the CMIP6 TCR, respectively. The gray and orange bars are the prior and posterior TCR ranges, respectively. The prior TCR range is defined by the primitive probability density estimate of the 20 TCR values without any weighting. The posterior TCR range is defined by the probability density estimate of the 20 TCR values weighted by the Kullback–Leibler divergences of the CMIP6 models. To derive the Kullback–Leibler divergence for each CMIP6 model, we apply the spread

of the 200 HadCRUT solar responses to each CMIP6 solar response and calculate $\Delta_i = \int_{-\infty}^{\infty} p(x) \ln\left(\frac{p(x)}{q(x)}\right) dx$, where $p(x)$ and $q(x)$ are the distributions of HadCRUT ensembles and the CMIP6 solar response, respectively. Note that the regression errors of the CMIP6 solar responses are much smaller than the HadCRUT spread, so a more conservative approach is to use the HadCRUT spread instead of using the regression errors. The weighting of each CMIP6 model is then given by $w_i = \frac{\exp(-\Delta_i)}{\sum_j \exp(-\Delta_j)}$.

Using Eq. (9), which is independent of CMIP6 models, we have

$$\overline{TCR} = a\,\mu_{SOLAR} = a\,b\,\kappa_{SOLAR} \qquad (12)$$

The proportionality constant, $b = \frac{\partial TSI}{\partial F_{SOLAR}}$, is the ratio of TSI and the ERF after stratospheric and tropospheric adjustment, where $\frac{\partial F_{SOLAR}}{\partial TSI} = \frac{1}{4}(1-\alpha)\xi$, $\alpha$ being the planetary albedo and $\xi$ the ratio of solar radiation at the top of the troposphere and top of the atmosphere.

The planetary albedo has been measured very accurately by satellites, with an accuracy of ±1% for the 2.5–97.5% range or,

equivalently, ±0.86 for the 5–95% range, with very little interannual variability and has a symmetry between the two hemispheres[38]. Gray et al. performed a wavelength dependent calculation of the albedo for the solar-cycle specific irradiation and concluded that it is almost the same as the wavelength-averaged planetary albedo. In contrast, the modeled planetary albedo is more variable, and perhaps overtly sensitive to the surface temperature. This explains some of the scatter of the slope in Fig. 6. Another cause for the scatter may be due to the slight differences in the solar-cycle response lags in models. We use observed values for these quantities to calibrate the emulator.

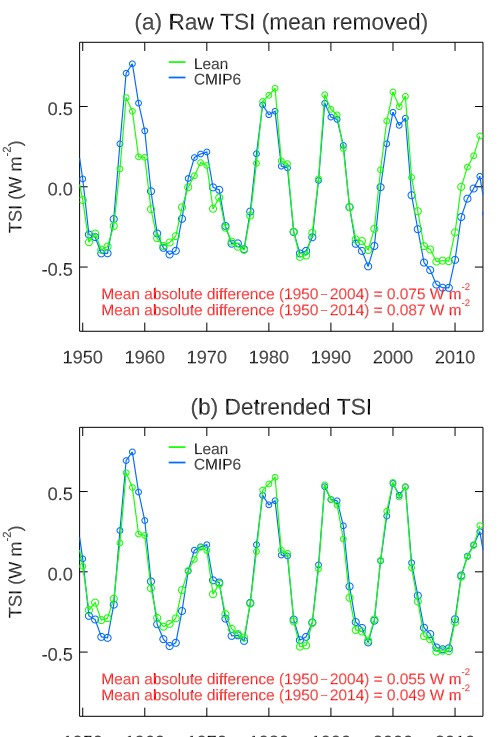

**Fig. 9 | Comparison of two reconstructed TSI time series. a** The two raw time series. **b** The detrended time series. The removal of the nonlinear secular trend is done using Empirical Mode Decomposition. The mean absolute difference is calculated by adding the absolute values of differences between the two time-series for each year, then the sum divided by the number of years.

The simulated value of $\xi$ is sensitive to the spectral resolution of the input solar spectrum due to the stratospheric $O_3$ absorption in the Ultra-Violet (UV) range. Gray et al.[39] estimated the stratosphere- adjusted radiative forcing (RF) for the 11-year solar cycle based on fixed dynamical heating approach and obtained a value of $\xi$ to be 0.78 using a solar UV spectrum with 1-nm resolution. -0.15 of the reduction from 1.00 is due to the stratospheric $O_3$ absorption of the solar variation at wavelength below 300 nm; the rest of the reduction is due to the combination of stratospheric $O_3$ absorption of the solar variation above 300 nm and the stratospheric temperature adjustment[40]. Earlier studies using coarser spectral UV resolutions obtained higher values: Larkin et al.[41] first obtained a value of 0.88 using a two-stream model with six spectral bands; Hansen et al.[42] reduced it to 0.83 using a solar spectrum at 5-nm resolution. We adopt Gray et al.'s[39] value with 1-nm resolution. AR6 also adopted this value for stratosphere-adjusted RF, to which it added −0.06 for tropospheric adjustment to yield $\xi = 0.72$ for use with ERF.

By imposing 5 satellite-observed solar-induced $O_3$ changes and using 2 radiative transfer schemes, Isaksen et al.[43] showed that the $O_3$ solar RF varied from −0.005 W m$^{-2}$ to 0.008 W m$^{-2}$ relative to the mean. These values are close to zero since an increase in ozone in the stratosphere absorbs more of the ultraviolet part of the solar radiation, but the increase in ozone heating emits more downward long-wave radiation that almost compensates the decrease in short-wave radiation. Larkin et al.[41] also tested their solar RF by replacing the simulated $O_3$ solar response with the observed, and the values remain the same as 0.23 W m$^{-2}$ in both cases due to the above-mentioned compensation. Because of their broadband calculations, they reported the resulting solar radiative forcing to only the second decimal place. We adopt Isaksen et al.'s values for the uncertainty in $\xi$. The standard deviation of the 10 net $O_3$ solar radiative forcings listed in Isaksen et al.'s[43] Table 1 is 0.0047. For a mean radiative forcing of $0.26 \times 0.72 = 0.187$ W m$^{-2}$, we

estimate a percentage uncertainty in $\xi$ to be $0.0047/0.187 \times 1.65$, which is ±4.1% (5–95% range). For observational data analysis we use: $\frac{\partial F_{SOLAR}}{\partial TSI} = \frac{1}{4}(1-\alpha)\xi = 0.127 \pm 4.2\%$, where $\xi = 0.72$ (±4.1%) (5–95% range), $\alpha = 0.29$ (± 0.86%) (5–95% range), and the total uncertainty for this quantity is $\sqrt{(4.1\%)^2 + (0.86\%)^2} = 4.2\%$ (5–95% range). The reciprocal is: $b = \frac{1}{0.127}(\pm 4.2\%) = 7.87 \pm 0.33$ (5–95% range).

The uncertainty in the reconstructed detrended TSI is 0.049 Wm$^{-2}$ for the period 1950–2014 (see "Methods" and Fig. 9), about 5%.

The solar response determined from HadCRUT5[17] yields, for the "very likely" range:

$$r_{TCR} = \sqrt{\left(\frac{\Delta \kappa_{SOLAR}}{\kappa_{SOLAR}}\right)^2 + \left(\frac{\Delta b}{b}\right)^2 + \left(\frac{\Delta a}{a}\right)^2 + \left(\frac{\Delta TSI}{TSI}\right)^2}$$
$$= \sqrt{\left(\frac{0.097-0.070}{0.084}\right)^2 + 0.084^2 + 0.086^2 + 0.05^2} \qquad (13)$$
$$= 34.2\%$$

Because some of the errors are not Gaussian, the above estimate may not be accurate. By performing Monte-Carlo bootstrapping of the slope, along with error in variable, we obtain the more accurate result of 34.4%, which is close to (13); both should be rounded to 34%. The best estimate for the normalized TCR is 0.66 °C/W m$^{-2}$.

For comparison, Padilla et al.[30] obtained estimates of TCR of 1.6 [1.3 to 2.6]°C, giving $r_{TCR} = 81\%$. They also used a two-layer emulator as here but with $T_d$ set to zero, and applied historical forcing and observed warming to constrain model parameters.

**TCR in conventional units**
Our central value in terms of normalized TCR, based on the Process-Based Estimate and the one-parameter Emergent Constraint, is 0.66–0.67 °C /Wm$^{-2}$. We take the average as the best estimate of the central value. For comparison with the TCR in conventional units, we need to multiply it by $F_{2 \times CO_2}$. The value of $F_{2 \times CO_2}$ to use, for consistency, should be what was divided into model TCR to normalize it as the vertical coordinate in Fig. 7. For the 5 models that are consistent with the observational constraints of solar amplitude and slope in these figures, the mean value is 3.38 W m$^{-2}$. Using this mean value, our best estimate (the average of process-based estimate and the emergent constraint) is TCR = 2.2 °C. Using AR6's mean value of 3.93 W m$^{-2}$ for doubling $CO_2$ forcing would have increased the value of estimated TCR in °C beyond AR6 TCR's "likely" and "very likely" ranges. However, this higher ERF value is not constrained by observations.

Using the common procedure of Emergent Constraint (with an intercept), a narrower range of:

$$2.2\,[2.0 \text{ to } 2.5]\,°C \qquad (14)$$

in conventional units is obtained here, which is 1/3 that of AR6's estimate using historical warming and Emergent Constraint. However, given our reservation about the 2-parameter fit used in the common Emergent Constraint procedure, a more conservative estimate is

$$2.2\,[2.0 \text{ to } 2.8]\,°C \qquad (15)$$

based on our best estimate of the central value and the range in Eq. (11).

**Sensitivity to including models that do not follow protocol**
There are 7 CMIP6 models that were excluded in Fig. 7 because they do not follow the CMIP6 protocol, which requires that models have at

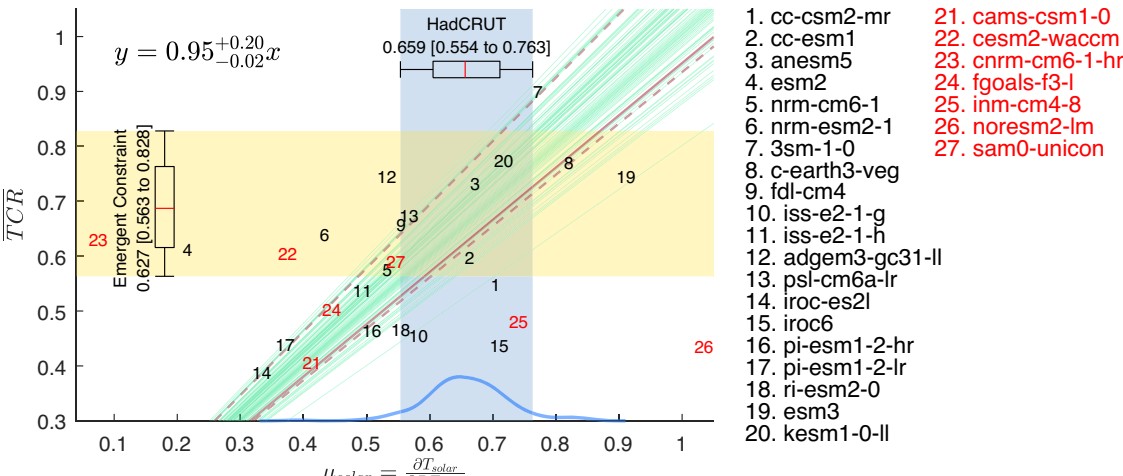

**Fig. 10 | Sensitivity to including more CMIP6 models.** Same as Fig. 7 except adding 7 models. These are models that do not satisfy the CMIP6 protocol of running the ocean spin up for over 500 years. CMIP6 model's normalized TCR (defined as $TCR/F_{2 \times CO_2}$) vs 11-year global-mean solar-cycle response $\mu_{SOLAR}$ in 2-m temperature regressed against the radiative forcing; both in units of °C/W m⁻². The shaded blue region indicates the spread of the observations. The regressed lines are drawn in green (only 50 are shown). The solid maroon line indicates the mean of the fit. The dashed maroon curves bound the 5–95% interval of the fit and should be interpreted as the prediction limits of the mean normalized TCR given an observed value of the solar response. The box and whisker represent the 17–83% and 5–95% "likely" and "very likely" intervals, respectively, obtained using 10,000 bootstrap samples of the 200 HadCRUT ensembles (each sample has 200 points with replacement), plus a Gaussian noise of $b$, which is the "error in variable". The super and subscripts indicate "very likely" interval of the estimated parameter.

least 500 years of spin-up of their oceans before the calculation of TCR. Including them (see Fig. 10) widens the uncertainty range slightly, from

$$\overline{TCR} = 0.67\,[0.58 \text{ to } 0.83]\,°\text{C/W m}^{-2}\,;\ r_{TCR} = 38\%$$

to

$$\overline{TCR} = 0.63\,[0.56 \text{ to } 0.83]\,°\text{C/W m}^{-2}\,;\ r_{TCR} = 43\%.$$

The upper range remains unchanged. It lowers the lower uncertainty range slightly from 0.58 to 0.56. We argue that these models should not be included.

## Discussion

Our observational constraint of Earth's Transient Climate Response based on the solar-cycle phenomenon is ~1/3 as narrow as existing estimates, which were previously all based on the observed historical warming. This reduction of uncertainty is achieved through a number of factors: (i) the solar phenomenon has more accurately measured forcing, compared to the large aerosol uncertainty in historical warming; (ii) the TCR from models are calculated using a consistent protocol; (iii) we use a climate sensitivity metric that involves normalizing model TCR by their individual radiative forcing, thus avoiding the including the latter's 12% uncertainty in TCR itself; (iii) only instrumental temperature record since 1950s is used.

With improvements in the modeling of physical processes, climate models have become more sensitive. CMIP6[1] models now generally have a higher TCR than previous generations, with a 5–95% inter-model range of 1.3 °C–3.0 °C. Our central estimate of 2.2 °C is near the midpoint of the intermodal spread of CMIP6 model TCRs, implying that some sensitive CMIP6 models (compared with previous generation) may be more consistent with the proposed observational constraint. In particular, climate models, such as GFDL-CM4, CANESM5, UKESM1-0-LL, BCC-ESM1 and IPSL-CM6A-LR, satisfy both of our observational constraints for TCR and for solar cycle response. We obtained essentially the same estimate without using CMIP6 models in a Process-Based Estimate. Our central estimate is

within AR6's[9] "very likely range" of 1.2 °C–2.4 °C. Our narrowing of the "very likely" range is mainly from a lifting, by more than 60%, of the low-end estimate.

Having an independent line of evidence in addition to the existing constraint based on historical warming potentially allows further reduction of the uncertainty by combining them. We leave this task to a future project.

Just as successive Assessment Reports yield more and more accurate estimates of the TCR as observational datasets improve, we expect that future estimates of TCR using the solar-cycle approach may need to be revised as surface temperature datasets change. We already saw changes when HadCRUT4 was replaced by HadCRUT5, which widened the uncertainty range and lowered slightly the mean. Also, as future generations of CMIP models include more solar cycles in the satellite era, accuracy of the extracted solar-cycle response is expected to improve; at this time we have to stop the comparison with observation at 2014. What we report here is our current best estimate.

## Methods
### Linear regression
Given two time-dependent datasets $x(t)$ and $y(t)$, we construct a linear regression model as: $y(t) = a\,x(t) + residue$. The linear regression coefficient $a$ is obtained from

$$a = \langle y(t)|x(t)\rangle = \frac{\int [x(t) - \bar{x}]\,[y(t) - \bar{y}]dt}{\int [x(t) - \bar{x}]^2 dt} = \frac{\sum_{i=1}^{n}(x_i - \bar{x})(y_i - \bar{y})}{\sum_{i=1}^{n}(x_i - \bar{x})^2} \quad (16)$$

which may be also viewed as the functional derivative $\frac{\partial y(t)}{\partial x(t)}$. The bars denote the temporal averages. The integration is over the length of the record. For sinusoidal forcing (e.g., the solar-cycle forcing), the integration is over a full period. For the TCR experiment it is over the 70 years of the experiment.

### Climate sensitivity and its uncertainties
The TCR is commonly defined by the difference between year 70 and year 1 global mean surface temperature. We use Δ to denote difference. So the normalized TCR is defined as the ratio of differences of

response temperature to forcing:

$$\overline{TCR} = \frac{\Delta T}{\Delta F}.$$

Note that $\Delta T$ and $\Delta F$ are not functions of $t$.

For transient climate sensitivity, the forcing is time-dependent, and so it is not appropriate to use the ratio of response to the forcing to define this sensitivity metric. A more appropriate measure of sensitivity to external forcing should be given as the regression of the response against the forcing. In the case of climate sensitivity, the response is taken to be the global mean surface temperature change ($T$) (after the removal of unforced variability), and the forcing is taken to be the global mean radiative forcing ($F$): $\mu = \langle T(t)|F(t)\rangle$, where $\mu$ is the climate sensitivity of interest. To obtain $\mu_{SOLAR}$, we constructed another regression coefficient: $\kappa_{SOLAR} = \langle T_{SOLAR}(t)|TSI(t)\rangle$, where $\kappa_{SOLAR}$ is the solar-cycle response we obtained using LDA (supplementary materials). We then obtained $\mu_{SOLAR}$ through the relation $\mu_{SOLAR} = b\,\kappa_{SOLAR}$, where $b^{-1} = \langle F_{SOLAR}(t)|TSI(t)\rangle = \frac{0.72(1-\alpha)}{4}$ and $\alpha$ is the planetary albedo.

It is difficult, though not impossible, to determine the equilibrium climate sensitivity from instrumental measurements because the current climate is far from equilibrium. The situation is different for TCR. Previously, the historical record of global warming change ($\Delta T$) in response to the change in radiative forcing ($\Delta F$) from the beginning of the industrial period to recent decades has been used, which is referred to as historical warming method or the Otto et al. method[6,8]:

$$\mu = \mu_{HIST} \sim \frac{\Delta T}{\Delta F}.$$

This ratio of differences is strictly speaking not a regression of response against forcing—perhaps we should have used a different symbol for it—but is close since the forcing and response are both approximately linear. The difference used in Otto et al. is the decade of 2000s minus 1860–1879. The best estimate for TCR is 1.3 °C, with a 5–95% range [0.9 °C to 2.0 °C]. Lewis and Curry[7], using the same method but different periods, also obtain 1.33 °C as the best estimate, but a wider spread of 0.9 °C to 2.5 °C, 120% of its central value, due mostly to their adopting AR5's $\Delta F$ with a wider uncertainty range. Their best estimate falls below the multi-model means of CMIP5, suggesting seemingly that some models overestimate future warming. This has generated much debate.

There are four major uncertainties in the historical warming method:

(i) As pointed out by Richardson et al.[8], "the single largest contribution to formal error in calculated TCR is, however, due to uncertainty in $\Delta F$". And the largest component of the uncertainty is due to aerosol forcing:[7] "Without a reduction in aerosol ERF uncertainty, additional observational data and extended time series may not lead to a major reduction in ECS and TCR estimation uncertainty". The spread of Otto et al.'s TCR estimates was mainly due to this large uncertainty in radiative forcing. Richardson et al.'s TCR has a spread of 1.0 °C to 3.3 °C, or 139% of its central value. This spread is even larger than that of Otto et al.'s TCR, not because of uncertainties in surface temperature records stated in (ii) below, but mostly because an updated, larger uncertainty for $\Delta F$ was used. This is where using the solar-cycle data has a marked advantage, as its forcing is much better measured and does not involve aerosol forcing.

(ii) A second source of error is in the historical data of surface temperature over such a long time span required in Otto et al.'s method, when earlier data are geographically less complete, and that data involved the blending of air temperature and sea

surface temperature, while models use surface air temperature in their TCR calculation. The adjustment in temperature record by Richardson et al. led to 24% higher warming: 15 of the 24 percentage points arise from masking models to HadCRUT4.4 geographical coverage and 9 from water-to-air adjustment in the model. This model-derived scaling factor was responsible for moving the "best estimate" from 1.33 °C to 1.66 °C, now within the CMIP5 model mean. For the solar cycle, we use only the modern record since 1950, aided also by the recent availability of a geographically complete dataset, GISTEMP Version 4, based on NOAA Global Historical Climatology Network (GHCN) Version 4 (meteorological stations) and ERSSTv5 (ocean)[44]. These geographically complete datasets yield a solar-cycle response very close to the central value of HadCRUT5.

For an assessment of the water-to-air temperature difference we use the 2-m temperature from European Center for Medium-Range Weather Forecasts (ECMWF) Re-Analysis (ERA). The 2-m temperature is obtained by interpolating the air temperature at the lowest level of the model (-10 m) and the skin temperature (SST), which in recent decades is remotely sensed. The latest ERA version 5 (ERA5) starts from 1979[45]. To extend the data before 1979, we combine the previous version, ERA-40, from 1960 to 1978 with ERA5 from 1979 to 2004[46]. The water-to-air temperature difference obtained from ERA is −10%. ERA uses a model to assimilate observed data, and so this adjustment can also be criticized as being model dependent. However, since this error is within the range of HadCRUT5's spread, we chose not to let it affect our reported result on TCR. In NCEP reanalysis the 2m-temperature is not a standard analyzed product.

(iii) A third source of uncertainty is related to the fact that the observed warming is the sum of forced response and unforced response. The latter includes the Atlantic Multidecadal Oscillation (AMO), believed to be caused by the Atlantic Meridional Overturning Circulation (AMOC)[47]. Tung et al.[48] and Chen and Tung[49] found that the large multidecadal variation in the observed global-mean surface temperature is mostly contributed by the AMO, and it can double the observed warming during its positive phase. Hu et al.[50] found that while AMO does not affect ECS, a weaker AMOC enhances surface warming and increases TCR (and vice versa). van der Werf and Dolman[51] found that the calculated TCR are affected substantially with different choice of AMO indices. Recent studies using post-1975 observed warming to constrain TCR[10,11] is even more prone to AMO contamination than using the centennial warming in Otto et al.'s method. In our analysis method, spatial-temporal information is used to extract the forced solar-cycle response, which greatly reduces the influence of AMO, which has a different spatial structure and much longer period.

## Sensitivity to aerosols from volcano eruptions

Volcanic aerosols tend to cool the surface and could affect our extracted solar-cycle signal, especially when the two large eruptions were spaced a decade apart. Two large volcanic eruptions, El Chichón in March 1982 and Pinatubo in June 1991, coincidentally erupted during solar max. So the response to the solar-cycle forcing could possibly be underestimated. However, the effect should be temporary and lasted two to three years. We tested the sensitivity of our method to the volcanic aerosols by excluding the two years after the El Chichón and also after Pinatubo eruptions. The resulting observed mean response and the 5–95% range are only 0.003 higher: 0.087 [0.073 to 0.100] respectively, in units of °C/W m$^{-2}$. The difference is well within the uncertainty of the HadCRUT5 data. The distribution of the HadCRUT5 ensemble responses with the $L$-block resampling test shown in Fig. 3 shows that the mean responses is still 99% confident.

## Extraction of solar-cycle signal

We use a spatial-time filter to extract the solar-cycle signal from the surface temperature dataset to minimize the contamination of the signal by other climate noises, such as El Niño. The same method is applied to both model and observations. The spatial part is the Linear Discriminant Analysis (LDA). The surface temperature data at each grid point is first linearly detrended before the application of LDA, to eliminate the possibility of global warming contaminating the solar signal, although this effect is small. We define a solar plus (minus) year as the year when the TSI is above (below) the local mean for the period under consideration. The local mean is found using the method of Empirical Mode Decomposition. Collectively the solar plus years are referred to as belonging to the "solar max group", and the solar minus years as belonging in the "solar min group". As described in more detail in the Supplementary Information, LDA finds the spatial pattern that best separates the solar plus years from the solar minus years given the criterion for defining these two groups. We do not use a threshold in defining the two groups and therefore no data are excluded for this purpose. The observed (or modeled) temperature field is projected onto this LDA pattern to produce a time series, denoted by $T_{LDA}(t)$. The second step of the procedure is to regress this time series onto the TSI, denoted by $S(t)$. This step further ensures that global warming signal is eliminated. It is the regressed solar-cycle response that is required in the definition of climate sensitivity, $\mu_{SOLAR}$.

## ERF for solar radiative forcing and its uncertainties

AR6 uses ERF (Effective Radiative Forcing) while earlier assessment reports used RF (Radiative Forcing). RF is defined as the radiative forcing at the tropopause after stratospheric ozone and circulation have adjusted, while ERF including fast tropospheric adjustment except surface temperature. The intent is to measure the effectiveness of radiative forcing in its effect in warming the surface in the same way as $CO_2$. ERF for solar radiative forcing is the TSI variation from solar min to solar max, divided by 4 (so that it is per unit area of a spherical earth), multiplied by the ratio $\xi$ of the downward radiation at the tropopause vs that at the top of the atmosphere, and multiplied by $(1 - albedo)$. The simulated value of $\xi$ is sensitive to the spectral resolution of the input solar spectrum due to the stratospheric $O_3$ absorption in the UV range. Gray et al.[39] estimated the adjusted RF for the 11-year solar cycle based on fixed dynamical heating approach and obtained a value of $\xi$ to be 0.78 using a solar UV spectrum with 1-nm resolution. -0.15 of the reduction from 1.00 is due to the stratospheric $O_3$ absorption of the solar variation at wavelength below 300 nm; the rest of the reduction is due to the combination of stratospheric $O_3$ absorption of the solar variation above 300 nm and the stratospheric temperature adjustment[40]. Earlier studies using coarser spectral UV resolutions obtained higher values: Larkin et al.[41] first obtained a value of 0.88 using a two-stream model with six spectral bands; Hansen et al.[42] reduced a value to 0.83 using a solar spectrum at 5-nm resolution. We adopt Gray et al's[39] value with 1-nm resolution. AR6 also adopted this value for Stratosphere-adjusted RF, to which it added −0.06 for tropospheric adjustment to yield $\xi = 0.72$ for use with ERF.

By imposing 5 satellite-observed solar-induced $O_3$ changes and using 2 radiative transfer schemes, Isaksen et al.[43] showed that the $O_3$ solar radiative forcing varied from −0.005 W m$^{-2}$ to 0.008 W m$^{-2}$ relative to the mean. These values are close to zero due to the fact that an increase in ozone in the stratosphere absorbs more of the ultraviolet part of the solar radiation, but the increase in ozone heating emits more downward long-wave radiation that almost compensates the decrease in short-wave radiation. Larkin et al.[41] also tested their solar radiative forcing by replacing the simulated $O_3$ solar response with the observed, and the values remain the same as 0.23 W m$^{-2}$ in both cases due to the above-mentioned compensation. Because of their broadband calculations, they reported the resulting solar radiative forcing to

only the second decimal place. We adopt Isaksen et al.'s values for the uncertainty in $\xi$. The standard deviation of the 10 net $O_3$ solar radiative forcings listed in Isaksen et al.'s[43] Table 1 is 0.0047. For a mean radiative forcing of $0.26 \times 0.72 = 0.187$ W m$^{-2}$, we estimate a percentage uncertainty in $\xi$ to be $0.0047/0.187 \times 1.65$, which is ±4.1% (5−95% range).

The planetary albedo has been measured very accurately by satellites, with an accuracy of ±1% for the 2.5−97.5% range or, equivalently, ±0.86 for the 5−95% range, with very little interannual variability and has a symmetry between the two hemispheres[38]. Gray et al.[39] performed a wavelength dependent calculation of the albedo for the solar-cycle specific irradiation and concluded that it is almost the same as the wavelength-averaged planetary albedo. In contrast, the modeled planetary albedo is more variable, and perhaps overtly sensitive to the surface temperature. This explains some of the scatter of the slope in Figs. [6], [7].

Thus, for observational data analysis we use:

$$\hat{F}_{SOLAR} = \frac{1}{4}(1 - \alpha)\,\xi = 0.127 \pm 4.2\% \tag{17}$$

where $\xi = 0.72 \pm 4.1\%$ (5−95% range), $\alpha = 0.29$ (±0.86%) (5−95% range), and the total uncertainty for $\hat{F}_{SOLAR}$ is $\sqrt{(4.1\%)^2 + (0.86\%)^2} = 4.2\%$ (5−95% range). The reciprocal is used in Eq. [5]: $b = \frac{1}{0.127} \pm 4.2\% = 7.87 \pm 0.33$ (5−95% range).

## Uncertainty in TSI reconstructions

Reconstruction of long-term variation of TSI is based on proxies of solar magnetic activity, mostly sunspot records and chromospheric indices on centennial time scale. A commonly accepted reconstruction is that by Naval Research Laboratory (NRL)[52]. Most CMIP6 used the recommendation by Mattes et al.[53]. It is difficult to provide an overall estimated of the uncertainties of these two reconstructions. We have decided to estimate the overall uncertainty of the TSI reconstruction by calculating the difference of the two datasets. The raw data are shown in Fig. [9]a, giving a mean absolute difference of 0.075 W m$^{-2}$ for the period 1950−2004 and 0.087 W m$^{-2}$ for the period 1950−2014. Much of the uncertainty is related to the secular trend, while our calculation uses only the difference between the solar max and solar min years. In Fig. [9]b we show the mean absolute difference of the detrended time series. Because the trend is nonlinear, linear detrending is not appropriate. We use the Empirical Mode Decomposition[18,19]. The mean uncertainty is 0.055 W m$^{-2}$ for the period 1950−2004. The mean difference is even smaller, at 0.049 W m$^{-2}$ for the period ending 1950−2014 because there are satellite measurements for the added 10 years. Since the TSI varies by about 1 W m$^{-2}$ from solar min to solar max, the uncertainty of 0.049 W m$^{-2}$ is about 5%. As can be seen in Fig. [9]b, the grouping of the solar plus years and solar minus years are identical with either of the solar reconstructions after nonlinear detrending, yielding identical solar temperature response using the LDA method.

## Uncertainty in method

The LDA spatial filter is very efficient in the sense that the projected time series $T_{LDA}(t)$ after the first step is already very close to the TSI time series[22], with a small residue of 0.02 °C−0.04 °C. This small residue is eliminated by the second, regression step. This residue in the intermediate step is not counted as an error in our two-step procedure.

The major uncertainty in the LDA method is related to the determination of the truncation parameter $r$ for the data regularization. $r$ should be large enough to include the target signal (the solar response in our case) but small enough to prevent overfitting with noise to generate an artificially high separation of the two groups. For HadCRUT5 over 1956−2014, the value of $\kappa$ reaches a plateau when

$r = 30$, implying that most of the solar response is included when $r \geq 30$. Further increasing $r$ leads to overfitting (Supplementary Information). Based on two different values of $r$ (30 and 38), we estimate an additional 1% to the overall uncertainty in $r_{TCR}$ in Eq. (13) for the 5–95% range for $\kappa$.

## Sensitivity to period considered

Solar forcing is not sinusoidal. Its TSI has variable amplitudes and slightly variable period. Different response to different forcing period is to be expected. This makes it more important that models and observations are analyzed using the same period when we want to use observations to constrain the models. When there is a secular trend in TSI, the nonlinear trend needs to be first removed. Here we use the Empirical Mode Decomposition[19,54] for that purpose. For the period 1950–2004, the secular solar trend is small; multi-cycle mean can be used instead. This is not always the case. Solar Cycle 24, from December 2008 to December 2019, is highly unusual. It is the lowest on record since Solar Cycles 12–15 during 1878–1923, leading many to speculate about the possibility of a "Grand Solar Minimum". We do not expect that our conclusion about model's TCR would be affected if both models and observations cover the same period, since we extract the response regressed against the forcing. However, including this extra cycle would include a secular trend, complicating the methods for defining solar plus years or solar minus years. This problem is solved using EMD to remove the nonlinear trend, and the two groups are defined relative to that nonlinear trend locally.

Amdur et al.[55] considered the period 1959–2019, including Solar Cycle 24, without removing the nonlinear trend. The year 1959 is a solar max and 2019 a min, which by themselves include an imbedded trend from solar max to solar min. They obtained $\kappa_{SOLAR} = 0.07 \pm 0.12\,°C/W\,m^{-2}$ (2.5–97.5% range) for HadCRUT4 when using the same LDA method as ours. (There may be a typo in their text, where they listed $0.10 \pm 0.12$, while their Table listed $0.07 \pm 0.12$). Their central value of 0.07 is very close to our 0.08, but their uncertainty range is more than twice that of their central value and includes 0, partly due to the secular trend in the solar forcing, and partly due to their phase randomization test, which likely led to an overestimated uncertainty, as discussed in Camp and Tung[25]. They further obtained different values when they slide the 60-year window to include different periods with unequal number of solar plus years and solar minus years. Having unequal number of the two solar groups introduces bias when we do not have sufficiently long record to average out the bias of having too many solar maxes or too many solar mins. There was also an error in their code when they were supposed to have used the Composite Mean Difference method to extract the solar cycle response: They actually used Composite "Sum" Difference, which amplifies the problem related to unequal number of samples in each group.

The signal extracted by other authors using purely time series analysis, such as multiple linear regression[56], tended to be smaller than what we found using space-time methods[20,57]. One should always use the most discriminating method known at the time. The difference between a more discriminating method and a less discriminating method should not be counted as an uncertainty in the method and has not been included in our uncertainty quantification. If this were counted, the uncertainty range can be arbitrarily large because there is always a worse method.

A method easier to implement than LDA is the Composite Mean Difference (CMD) method of Camp and Tung[25], and the associated statistical tests as described in detail in Zhou and Tung[20]. CMD uses the difference in spatial structure between the mean of solar plus years and solar minus years and projecting the space-time data onto that spatial pattern. CMD is more intuitive and yields the solar cycle signal that is very close to that obtained by LDA but has a larger uncertainty. For this reason, CMD is not employed in this study.

## Uncertainty in RF for $CO_2$

Myhre et al.[58], obtained by fitting to the results of a broadband model, the following canonical formula for the RF of 2×$CO_2$:

$$F_{2 \times CO_2} = 5.35 \ln\left(\frac{C}{C_0}\right) W\,m^{-2},$$

where $C$ is the atmospheric concentration of $CO_2$ and $C_0$ is its initial value. For doubling $CO_2$, this formula yields $F_{2 \times CO_2} = 3.71\,W\,m^{-2}$. The constant coefficient has an uncertainty of 1%. The broadband model has a reported error of −2.4% compared to the line-by-line model. Updating the line-by-line radiative transfer calculations and after reviewing the uncertainties involved, Etminan et al.[59] concluded that the total uncertainty for $F_{2 \times CO_2}$ is ±10% (5–95% range), retaining what was previously adopted by IPCC AR5. IPCC AR6 found ERF to be $0.2\,W\,m^{-2}$ higher than Stratosphere adjusted RF, with a ±12% uncertainty. We suggest that this uncertainty in model diagnosed ERF, which is not an observable quantity, can be avoided by constraining the normalized TCR. In climate model runs, this ERF does not need to be specified. The doubling $CO_2$ atmospheric concentration can be specified without uncertainty.

## Uncertainty in surface temperature data

HadCRUT5[17] is a combination of a global land-surface temperature dataset, CRUTEM5, and the global SST dataset, HadSST4. There is no interpolation (in-filling) performed and so there are some coverage gaps. Although poor in the 1850–1880 period, the data coverage is good after 1950, at over 75%. We use only the data after 1950. Two hundred ensemble members, an increase from the 100 ensemble members in the previous version, are used to provide an assessment of observational uncertainties, such as those arising from changing instrumental and observational practices, changes in station locations, and changes in local land use. These uncertainties are shared by other datasets. We apply the LDA method to each ensemble member to extract a solar-cycle response. The larger number of ensemble members yields a larger uncertainty range (about twice) compared from that obtained from the previous version, HadCRUT4.6. The central value is obtained as the ensemble mean.

Richardson et al[8]. pointed out the importance of using a geographically complete dataset. They found a significant 15% difference in the inferred TCR between a masked version and the complete model dataset and adjusted their mean value of TCR by this modeled ratio to account for the effect if the dataset were complete. However, this adjustment is model dependent. Currently one geographically complete observational dataset is available, provided by GISTEMP, which builds upon the latest NOAA Merged Land–Ocean Global Surface Temperature Analysis. The NOAA sea-surface temperature (ERSSTv5) is complete, based on satellite observations since 1979, but some missing data exist over land. Unlike HadCRUT5, GISTEMP does not provide a way for us to assess their uncertainty. Its solar-cycle responses however fall within the uncertainty range of HadCRUT5 (see Fig. 1).

In 2008, because geographically complete datasets were not available, Tung et al.[22] included two reanalyses, ERA-40 and NCEP. Solar-cycle response from ERA-40 and NCEP differ significantly. While ERA-40's solar-cycle response is 0.12 °C, NCEP is an outlier at 0.17 °C. While ERA-40 produced 2-m temperature by interpolating from SST, NCEP's "surface" temperature (at 0.995 sigma level) was produced by the model using upper air observations and surface pressure. It does not assimilate the observed 2-m temperature over land[60]. Tung and Camp[61] found the solar signal obtained from NCEP's "surface" temperature to be the same as that found from its 925-hPa temperature, which tends to be higher in magnitude than that from the surface temperature. NCEP's interannual variance is generally higher than ERA's[61]. It should be pointed out that the NCEP's higher value was not

used in constraining the TCR in Tung et al.[22]. At the time, the linear relationship in Eq. (12) was not available because the reported radiative transfer calculations available (from the top of the atmosphere to the tropopause) were to us ambiguous. That study resorted to an inequality that the TCR should be larger than a minimum value corresponding to the lowest solar-cycle response. NCEP's higher value was therefore not used as a constraint. It was used only to indicate the uncertainty of this lower bound and did not affect the conclusion. HadCRUT3 used then has now been updated to HadCRUT5 and GISS has gone through a major update incorporating NOAA's ERSSTv5. The newer datasets are geographically more complete, and lead to lower solar-cycle responses in surface temperature.

## CMIP6 models

Previously, it was thought that climate models either cannot generate solar-cycle signals at the surface or, if they could, the amplitude would be too small to be detected over climate noise[62]. Consequently, the solar-cycle response has not been used to constrain model climate sensitivities. Some still think the surface response to the small radiative forcing is too small as calculated from a back-of-envelope estimate, without realizing that, like greenhouse gas warming, the response should be amplified by the various climate feedbacks in the troposphere. It was also argued by some studies that exotic mechanisms not included in CMIP models, such as cosmic rays forming condensation nuclei for clouds, need to be invoked. Solar-cycle responses from the climate models are extracted from each model's surface temperature using the method of LDA[61,63], which utilizes the spatial-temporal information of the phenomenon. The models studied here include some with interactive ozone chemistry and some without, and yet there is no systematic difference between them in their solar-cycle response. This is noteworthy as one of the proposed mechanisms, the so-called "stratospheric pathway"[64], involves interactive ozone in the stratosphere: The UV portion of the solar irradiance produces ozone in the stratosphere, and the additional ozone heating generates a stratosphere temperature signal. Although it is known that very little of the ultraviolet portion of the 11-year solar cycle forcing gets through the stratosphere into the troposphere because of the effective ozone absorption in the stratosphere, there have been some proposals for an indirect influence of the stratosphere on the troposphere[64]. The proposed pathway may affect regional variability for which an interactive ozone model is important. Here we study the global surface signal.

There have been proposals of the galactic cosmic rays (GCR) affecting the troposphere via their forming condensation nuclei for clouds. In fact, it was suggested that the early twentieth century global warming was caused by the cosmic rays. The second author, Tung, was in a National Academy of Sciences team that examined this issue in 2012[62] and found no evidence supporting the proposal and evidence contradicting it. Later in 2017 a satellite called "CLOUD" was launched to study this issue. The results were that the cosmic rays are "unlikely to be comparable to the effect of large variations in natural primary aerosol emissions".

Fluxes of energetic particles consisting of corona mass ejection of energetic particles (mostly ionized hydrogen) and the effect of solar wind on cosmic rays can potentially affect atmospheric composition, but mostly in the upper atmosphere. Most CMIP6 models are unable to account for these particle fluxes with the exception of a few, such as WACCM. Notable recent studies focusing on the impacts of GCRs, solar proton events and energetic electron precipitation on chemical composition of the atmosphere include Calisto et al.[65] and Rozanov et al.[66]. They showed using the SOCOL model that the GCR induced changes in the chemical species, such as $NO_x$, $HO_x$, $O_3$, are largest as measured in percentage (~10%) in the middle troposphere. However, since these chemical tracers are most abundant in the stratosphere and less abundant in the troposphere by at least an order of magnitude, the contribution of the tropospheric change to the total $NO_x$ abundance is very small.

In our recent publications[67,68] we estimated the solar-cycle induced changes in $NO_2$ using ground-based observations, WACCM, and our own 1-D photochemical model. The changes are largely in the stratosphere, near the 1-hPa level, with an exception during rare non-linear events such as tropopause folding in the mid-latitude[68]. Thus, the overall solar-cycle (or GCR) induced changes in the tropospheric $NO_x$ is much less than other variabilities, e.g. anthropogenic pollution, and therefore we do not focus on its climate impact in the current study.

The aforementioned studies also suggested that the GCR-induced changes of stratospheric species could lead to a reduction in stratospheric $O_3$, which in turn could strengthen the polar night jets and potentially indirectly impact surface climate [e.g. in northern Europe]. Matthes et al.[53] used one of the CMIP6 model, WACCM and obtained a weakened polar night jet in the solar maximum, consistent with those studies, potentially impacting regional climate. We are interested however in globally averaged surface response.

In summary, these and the studies mentioned earlier suggest that while these particle fluxes may affect chemical composition in the upper atmosphere, their impact on surface global temperature response is likely small. Furthermore, even if the effect is not small, it does not affect our results as long as these variations follow the time variation of the $TSI(t)$, though some may be anticorrelated. This is because we extract the total solar cycle response from all solar forcing and regressed against the $TSI(t)$ for both models and observation. In observation, obviously the effect is included. In the few models which include particle flux forcing and SSI, the response should include it. Though if such particle forcing and or SSI is important, the denominator of the regression could be larger or smaller. But since both the models and observations used the same regression, the relationship between them does not change.

## CMIP6 models and their TCR

Model TCRs were taken from Meehl et al.[1] for CMIP6. Uncertainty in how TCR is calculated is usually not adequately included in previous uncertainty analyses of climate sensitivity. As pointed out in our previous work[69], one major uncertainty is the internal variability in the control run affecting the baseline temperature which was subtracted from the 70th year temperature to calculate TCR. This variability yields TCR values that could be 30% different for the same model depending on which year is chosen for the baseline. We previously recommended taking a 100-year mean of the preindustrial control run for the baseline value, and that the control run be at least 1,000 years long. We also showed that the subtraction of the climate drift of a parallel control run is no substitute for adequately long control runs. These protocols are partially adopted for the CMIP6 protocol: The control runs were recommended to be at least 500 years, and the ESMValTool in Meehl et al. calculated TCR using 20-year averaging of the 60th–79th years of the run minus the Pre-Industrial baseline value. For some models the control runs are still too short, and substantial uncertainty may remain[69,70]. In AR4, the model with the highest TCR was MIROC(hires), at 2.6 °C. It had only 109 years of coupled spin-up. MIROC6 now has 1000 years of ocean spin up. The MIROC-ES2L model has 3,000 years of ocean spin-up and 1000 years of coupling to the atmosphere. Their TCR of 1.6 is now among the lowest in CMIP6. The uncertainty associated with inadequate time for the spin-up and control runs remains unquantified and unquantifiable by us. Therefore, we exclude from our study models with such problems (See Supplementary Table S1).

Solar responses simulated in all ensemble members of the same model are averaged to give a single value that is listed in Supplementary Table S1, except for MIROC6. MIROC6 has 50 ensemble members starting with different branch-off times of the pre-industrial simulation. Some members branched off too early, not giving adequate

spin-up time before branching. The average solar response of MIROC6 is here obtained using the 20 ensemble members that have branch-off times of at least 600 years.

### Lag in solar-cycle response relative to forcing
The lag extracted from observations is close to zero. Since we used annual mean data to avoid the rather large seasonal variability, a lag of a few months cannot be ruled out, but such a lag cannot be more than six months. We assign an uncertainty in this slope using the two extreme values of lag, 0 and 6 months.

## Data availability
All data used in this study are publicly available. The CMIP6 data were downloaded from https://esgf-node.llnl.gov/. The ERA-40 and ERA5 data were downloaded from https://www.ecmwf.int/. The HadCRUT5 data were downloaded from https://www.metoffice.gov.uk/hadobs/hadcrut5/. The GISTEMP4 data were downloaded from https://data.giss.nasa.gov/gistemp/. The solar flux used in CMIP6 models can be obtained at https://solarisheppa.geomar.de/cmip6. This is also the solar forcing used here in our analyses. For comparison purposes we also used the Naval Research Lab's solar forcing: https://data.giss.nasa.gov/modelforce/solar.irradiance/

## Code availability
The algorithm of the linear discriminant analysis (LDA) used to extract the solar signal in this study is contained in the Supplementary Information. The work presented in the main text was obtained using the LDA algorithm implemented in IDL.

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

## Acknowledgements

KFL has been supported in part by the National Science Foundation's Coupling, Energetics, and Dynamics of Atmospheric Regions Program under Grant 2230265. KKT's research was supported in part by National Science Foundation under Grant 1536175, and by the Frederic and Julia Wan Endowed Professorship.

## Author contributions

K.-F.L. and K.-K.T. conceived the project and designed the experiments. K.-F.L. carried out the experiments, performed the analysis and statistical tests. K.-.F.L. developed the tools and wrote the Supplementary Information. K.-K.T. interpreted the results and wrote the main parts of the paper.

## Competing interests

The authors declare no competing interests.

## Additional information

**Peer review information** *: Nature Communications* thanks Diego Jiménez de la Cuesta and the other, anonymous, reviewers for their contribution to the peer review of this work. A peer review file is available.

