## [Peer Review File · Nature Communications]

Solar Cycle as a Distinct Line of Evidence Constraining Earth's Transient Climate ResponseREVIEWER COMMENTS

Reviewer #1 (Remarks to the Author):

The authors use global annual mean temperature response to the 11 year solar cycle to find a relationship with TCR in a (limited) set of 20 CMIP6 models and then use this to constrain TCR with observations.

The idea is a nice one, although I am surprised it works as well as the authors claim – the amplitude of the solar cycle is ~ 0.2 W/m² which is very small compared to other forcing factors and internal variability.

Regarding the para starting line 307: The result would be more convincing if more CMIP6 models were added. Adding the models with reduced piControl run length should not (in principle) affect the result if it is robust. It would be good to try this with the CMIP5 models too.

I have further doubts based on figure 3 – the cluster of numbers representing each of the models does not look like it has a good correlation. What is the Pearson correlation coefficient here? Same with figure 4.

Reviewer #2 (Remarks to the Author):

I think that the main idea of this article is of great value. For ECS, the radiative feedback mechanisms and the ocean's thermal energy distribution had an enormous influence on the energy budget; therefore, they are the key point for reducing the uncertainties.

In contrast for TCR, the transient changes in the forcing have more relevance in comparison to the radiative feedback mechanisms and the ocean's energy uptake. Therefore, uncertainties in the forcing become more important for understanding how the system responds.

The authors of this article propose that we need another source of information for TCR, as we are depending too much on the surface temperature in the historical period, to which the models have been tuned. Therefore, the authors look for an alternative line of evidence.

Since the forcing is more relevant for estimating TCR, then looking at how the Earth system reacts to different types of radiative forcing is of great value, even though the other forcing mechanisms can activate the radiative feedback mechanisms in a different way.

General evaluation:

I think that the article is fit for publication in the present state except for some minor to medium details which include some clarifications about assumptions and notation issues. I summarise the clarifications in the commented manuscript. However, I think that the article is entirely understandable, the methodology sound and the article will have a high impact on the climate sensitivity community. For those reasons, I recommend publication after the clarification and possible rewriting.

Reviewer #3 (Remarks to the Author):

The manuscript entitled “Solar cycles as a distinct line of evidence constraining Earth’s transient climate response” by Li and Tung describes a study concerning Transient Climate Response (TCR) estimates of climate models. The study uses several quantities and methods that exploit the solar-cycle forcing to constrain TCR estimates of climate models.

The solar activity, whose primary manifestation is the approximately eleven years variation of sunspots emerging onto the Sun's surface, also characterizes the electromagnetic and particle emissions of the Sun. The Earth’s climate response to the Sun’s output variations depends on the direct effects of changes in the total solar irradiance (TSI) and solar spectral irradiance (SSI) on atmospheric and surface heating rates, and on the indirect effects of SSI fluctuations and of variations in the fluxes of energetic particles on atmospheric composition. TSI, SSI, and energetic particles' fluxes are found to vary at all time scales covered by direct measurements, and they are expected to vary also on longer time scales. For many years, climate models only accounted for TSI changes, but recent models have started to cover all the relevant solar irradiance and energetic particle contributions with increasing accuracy. Severity of Earth’s warming predicted by climate models depends on their TCR. Reducing the range of uncertainty for TCR of climate models has a significant economic value.

The manuscript consists of a main text, a section describing the methods applied and supplementary material presenting the technique employed to extract the solar-cycle response from the HadCRUT5 surface temperature dataset and main specifications of the CMIP6 models considered in the study. The content of the manuscript suggests that the authors have analyzed several datasets and worked out various issues to achieve the main result of their study, which is a reduction of the range of TCR estimates derived from climate models by adopting a new metric proposed by the authors. However, we wish to express our concern that this result derives from analytical and numerical modelling of data obtained from observations and from climate models under several assumptions. Thus, it does not contribute to actual improvement of the current knowledge of warming predicted by climate models that ultimately depends on relevant physics in play entered in the models in compliance with observational data. In addition, the manuscript and study are unclear in several respects. Focusing only on the solar forcing considered by the authors, we remark that the manuscript does not provide the reader with basic information needed to review and replicate the study. Indeed, the TSI dataset utilized by the authors is not even specified. Besides, several arguments concerning solar forcing, although correct, do not match the data and methods employed in the study. For example, the authors often refer

to the high accuracy of top-of-the-atmosphere solar-cycle forcing measured by satellites, but the data considered in their study cover the period 1951-2004, which includes the interval 1951-1978 uncovered by TSI satellite measurements. Thus roughly half of the data employed do not support the authors' claim. Besides, the data considered by the authors omits the TSI measurements recorded since 2004 without solid justifications. As for solar cycle 24, not covered by the study, we wish to point out that it was unusual but not unique and thus worth considering when dealing with climate response. Moreover, while TSI monitoring revealed clear variations in phase with the solar activity cycle by roughly 0.1%, i.e. close to the 1 Wm^{-2} changes reported in the manuscript, it also showed TSI fluctuations with amplitudes of up to 0.3% on the time scales of days to weeks, which are 3 times larger than the ones observed on decadal time scales. The short-term TSI fluctuations are unconsidered in the study, as are the TSI changes on time scales longer than the solar cycle, the variations of SSI at diverse spectral bands, and the fluctuations of particles fluxes, all these terms being relevant to climate studies. Indeed, the effect of some of these terms are actually reported in the manuscript, e.g. the increased uncertainty due to inclusion of a secular trend, and impact of UV SSI changes on O3 and albedo. However, the authors' claims on these terms refer to other studies, and are not supported by the evidence of arguments and data derived from the experimental period considered by the authors. In addition, we note that minima and maxima of solar cycles are usually defined differently to how the authors did, with potential effects on the results achieved by the study that seem unexplored by the authors. Finally, we wish to express our concern that overall the manuscript is difficult to read due to statements that are unclear, incomplete, or not thoroughly justified.

Based on the above problems, which are only a few examples relevant to solar forcing, I do not recommend the manuscript for publication in its present form.

REPLY TO REVIEWER #1

Reviewer #1

The authors use global annual mean temperature response to the 11 year solar cycle to find a relationship with TCR in a (limited) set of 20 CMIP6 models and then use this to constrain TCR with observations.

The idea is a nice one, although I am surprised it works as well as the authors claim – the amplitude of the solar cycle is ~ 0.2 W/m² which is very small compared to other forcing factors and internal variability.

We thank Reviewer #1 for the comments. We understand the Reviewer's concern and have provided more discussion in the revised manuscript to address it. Evidence supplied are: (1) if there is no solar-cycle forcing, could our method still get a "solar cycle response" due to contamination from other internal variability or forcing? The answer is no. Please see Figure S1. (2) If there is a simultaneous presence of solar cycle forcing and other internal variability, can the extracted solar cycle "response" be possibly contaminated by the other internal variability? Most of the CMIP6 models have multiple ensemble runs. Since the ensemble mean is close to the forced solution, our use of the ensemble means largely removed the internal variability which may be present in individual members. However, the number of members is small and so this step cannot guarantee the elimination of the internal variability entirely. Our spatial-time method has two steps. After the second step (regression against TSI), internal variability is much reduced. (3) Since the solar cycle response and global warming have similar spatial pattern, could our method include some of the global warming signal as solar cycle? This is unlikely, since the surface temperature is first detrended, and then in the second step of our method the signal is further regressed against a sinusoidal time series (the TSI). Global warming is not sinusoidal.

Regarding the para starting line 307: The result would be more convincing if more CMIP6 models were added. Adding the models with reduced piControl run length should not (in principle) affect the result if it is robust. It would be good to try this with the CMIP5 models too.

We have now followed the Reviewer's suggestion and included all 27 CMIP6 models. The figure is shown as Figure 7 in the main text. Although our conclusions remain little changed, we believe the possibility exists that this is just a coincidence. In our previous publications (Liang et al, 2013), we showed that if the piControl run lengths are shorter than 1000 years, the TCR calculated could vary by up to 30% depending on which year was taken as the initial year for the calculation. Therefore we do not believe that the results with short pi-Control runs are robust. For example one of the 27 models, WACCM, has only 133 years of spin up, and the large variability caused by the model adjustment to the initial shock overwhelms the TCR and solar cycle signal.

CMIP5 models do not have a protocol for calculating TCR, and so they are subject to the criticism of Liang et al (2013).

I have further doubts based on figure 3 – the cluster of numbers representing each of the models does not look like it has a good correlation. What is the Pearson correlation coefficient here? Same with figure 4.

The Pearson correlation coefficient for the cluster of points in Figure 3 (now Figure 4) is 0.7 ($R^2=0.49$) and it is statistically significant. This information is added to the figure. For Figure 4 (now Figure 5) it is 0.98 ($R^2=0.97$). We should point out that this very high value is due to the fact that the line is constrained to pass through the origin. This caveat is included in the revised manuscript.

REPLY TO REVIEWER #2

Reviewer #2 (Remarks to the Author):

I think that the main idea of this article is of great value. For ECS, the radiative feedback mechanisms and the ocean's thermal energy distribution had an enormous influence on the energy budget; therefore, they are the key point for reducing the uncertainties.

In contrast for TCR, the transient changes in the forcing have more relevance in comparison to the radiative feedback mechanisms and the ocean's energy uptake. Therefore, uncertainties in the forcing become more important for understanding how the system responds.

The authors of this article propose that we need another source of information for TCR, as we are depending too much on the surface temperature in the historical period, to which the models have been tuned. Therefore, the authors look for an alternative line of evidence.

Since the forcing is more relevant for estimating TCR, then looking at how the Earth system reacts to different types of radiative forcing is of great value, even though the other forcing mechanisms can activate the radiative feedback mechanisms in a different way.

We thank Reviewer #2 for his/her very positive comments and insights on our approach, which were expressed more succinctly than we could ourselves.

General evaluation:

I think that the article is fit for publication in the present state except for some minor to medium details which include some clarifications about assumptions and notation issues. I summarise the clarifications in the commented manuscript. However, I think that the article is entirely understandable, the methodology sound and the article will have a high impact on the climate sensitivity community. For those reasons, I recommend publication after the clarification and possible rewriting.

We thank Reviewer #2 for the very positive recommendation. We appreciated the detailed comments provided in his/her tracked manuscript. We have addressed each one as Replies to the Comment in the tracked manuscript, which is attached. Revisions are done to address the issues raised.

We have accepted the Reviewer's suggestions in the comments in the tracked changes about rewriting several sentences to avoid confusion to readers. This is done in the revised version of our manuscript.

There were several comments in the tracked manuscript about our notation for regression. We have now changed the confusing notation. TCR was defined using the

difference between the global temperatures at year 70 vs year 0 using the Δ notation, as in:

$\frac{TCR}{\Delta F} = \frac{\Delta T}{\Delta F}$. Our metric was a differential one and was previously denoted by

$\mu = \frac{\partial T(t)}{\partial F(t)}$. It can be viewed as a functional derivative, or as a partial variation in the

regression analysis. Since $F(t)$ is linear in time, this metric measures the linear-in-time part of the temperature response. To avoid confusion with the usual partial derivative, now we changed the notation to

$\mu = \langle T|F \rangle$.

The Reviewer would like to see a comparison between the spatial patterns of solar cycle and CO₂-induced warming in the latest generation of climate models. It is provided as Figure 2 in the revised version of the manuscript.

¹REPLY TO REVIEWER #3

Reviewer #3

The manuscript entitled “Solar cycles as a distinct line of evidence constraining Earth’s transient climate response” by Li and Tung describes a study concerning Transient Climate Response (TCR) estimates of climate models. The study uses several quantities and methods that exploit the solar-cycle forcing to constrain TCR estimates of climate models.

The solar activity, whose primary manifestation is the approximately eleven years variation of sunspots emerging onto the Sun’s surface, also characterizes the electromagnetic and particle emissions of the Sun. The Earth’s climate response to the Sun’s output variations depends on the direct effects of changes in the total solar irradiance (TSI) and solar spectral irradiance (SSI) on atmospheric and surface heating rates, and on the indirect effects of SSI fluctuations and of variations in the fluxes of energetic particles on atmospheric composition. TSI, SSI, and energetic particles’ fluxes are found to vary at all time scales covered by direct measurements, and they are expected to vary also on longer time scales. For many years, climate models only accounted for TSI changes, but recent models have started to cover all the relevant solar irradiance and energetic particle contributions with increasing accuracy. Severity of Earth’s warming predicted by climate models depends on their TCR. Reducing the range of uncertainty for TCR of climate models has a significant economic value.

The manuscript consists of a main text, a section describing the methods applied and supplementary material presenting the technique employed to extract the solar-cycle response from the HadCRUT5 surface temperature dataset and main specifications of the CMIP6 models considered in the study. The content of the manuscript suggests that the authors have analyzed several datasets and worked out various issues to achieve the main result of their study, which is a reduction of the range of TCR estimates derived from climate models by adopting a new metric proposed by the authors. However, we wish to express our concern that this result derives from analytical and numerical modelling of data obtained from observations and from climate models under several assumptions. Thus, it does not contribute to actual improvement of the current knowledge of warming predicted by climate models that ultimately depends on relevant physics in play entered in the models in compliance with observational data. In addition, the manuscript and study are unclear in several respects. Focusing only on the solar forcing considered by the authors, we remark that the manuscript does not provide the reader with basic information needed to review and replicate the study. Indeed, the TSI dataset utilized by the authors is not even specified. Besides, several arguments concerning solar forcing, although correct, do not match the data and methods employed in the study. For example, the authors often refer to the high accuracy of top-of-the-atmosphere solar-cycle forcing measured by satellites, but the data considered in their study cover the period 1951-2004, which includes the interval 1951-1978 uncovered by TSI satellite measurements. Thus roughly half of the data employed do not support the authors’ claim. Besides, the data considered by the authors omits the TSI measurements recorded since 2004 without solid justifications. As for solar cycle

24, not covered by the study, we wish to point out that it was unusual but not unique and thus worth considering when dealing with climate response. Moreover, while TSI monitoring revealed clear variations in phase with the solar activity cycle by roughly 0.1%, i.e. close to the 1 Wm^{-2} changes reported in the manuscript, it also showed TSI fluctuations with amplitudes of up to 0.3% on the time scales of days to weeks, which are 3 times larger than the ones observed on decadal time scales. The short-term TSI fluctuations are unconsidered in the study, as are the TSI changes on time scales longer than the solar cycle, the variations of SSI at diverse spectral bands, and the fluctuations of particles fluxes, all these terms being relevant to climate studies. Indeed, the effect of some of these terms are actually reported in the manuscript, e.g. the increased uncertainty due to inclusion of a secular trend, and impact of UV SSI changes on O3 and albedo. However, the authors' claims on these terms refer to other studies, and are not supported by the evidence of arguments and data derived from the experimental period considered by the authors. In addition, we note that minima and maxima of solar cycles are usually defined differently to how the authors did, with potential effects on the results achieved by the study that seem unexplored by the authors. Finally, we wish to express our concern that overall the manuscript is difficult to read due to statements that are unclear, incomplete, or not thoroughly justified. Based on the above problems, which are only a few examples relevant to solar forcing, I do not recommend the manuscript for publication in its present form.

We thank Reviewer #3 for the many comments related to solar **forcing**. Our original presentation was admittedly weak in this area as we focused our attention on the **response** to the forcing. So the Reviewer's comments help us strengthen this part of our presentation.

- (1) We neglected to specify the solar forcing dataset used in our work. This is now specified. We thank the Reviewer for pointing out our oversight.
- (2) *Besides, several arguments concerning solar forcing, although correct, do not match the data and methods employed in the study. For example, the authors often refer to the high accuracy of top-of-the-atmosphere solar-cycle forcing measured by satellites, but the data considered in their study cover the period 1951-2004, which includes the interval 1951-1978 uncovered by TSI satellite measurements.*

We accept this criticism. We previously omitted in our discussion the reconstructed solar forcing for the period of 1951-1978. This is now included. A new figure, Figure S6, is used to give an estimate of the uncertainty in TSI from 1950 to 2014.

- (3) *The short-term TSI fluctuations are unconsidered in the study, as are the TSI changes on time scales longer than the solar cycle, the variations of SSI at diverse spectral bands, and the fluctuations of particles fluxes, all these terms being relevant to climate studies.*

As mentioned by the Reviewer, the TSI has 0.3% fluctuations on the time scale of days to weeks. We did not consider these short time scale fluctuations, as we considered

only annually averaged TSI over the decadal time scale. We are interested in climate responses and not interested in the daily or monthly fluctuations. Is it justified? We think so, since the forcing is small the response is likely linear, and so we should be able to treat the longer time scale response separately from the short time scale ones. Our study uses the response to the relative variation between solar max and solar min to constrain climate sensitivity. TCR, as defined, depends only on decadal variations in radiative forcing. This is our goal. On longer time scales, scales longer than 11 years, one may want to use the 22-year cycle to constrain the TCR but the observation is too short for us to know what the actual observed response is. We did not use the secular trend of solar forcing to constrain TCR because it is more uncertain and there is no need to do so.

- (4) Although our use of the terms “solar max” and “solar min” throughout the text is consistent with common usage, there are some offending lines in the Methods dealing with the separation of the data into two groups for the LDA method to maximize the distance between them: “We define a solar max (min) year as the year when the TSI is above (below) the multi-cycle mean for the period under consideration.” We agree with the Reviewer that the names of the two groups are not the common usage. We have now changed them to the “solar plus” and “solar minus” groups, to denote that they are in the positive and negative phases of the solar cycle, respectively.
- (5) *Besides, the data considered by the authors omits the TSI measurements recorded since 2004 without solid justifications. As for solar cycle 24, not covered by the study, we wish to point out that it was unusual but not unique and thus worth considering when dealing with climate response.*

We previously discussed why we did not include solar cycle 24. The reason is more technical than physical. Solar cycle 24 is unusual in that it has more solar minus years than solar plus years since it has a long flat solar minimum and a short solar maximum. When there are more solar minus years than solar plus years it causes technical problems with our method. We have now included solar cycle 24 in our calculation. Please see the new Figure 5.

- (6) *Thus, it does not contribute to actual improvement of the current knowledge of warming predicted by climate models that ultimately depends on relevant physics in play entered in the models in compliance with observational data.*

Although the latest climate models incorporate as much relevant physics that is understood, they also have subgrid parameterizations that are not based on first physical principles but contribute significantly to cloud feedback and water vapor feedback, which ultimately determine the magnitude of the predicted warming. The purpose of our work is to find a way to decide which model has a climate gain to radiative forcing that is consistent with observation and which model has an inconsistent one. It contributes to our knowledge about warming predicted by climate models by informing us of which model is “too hot” and which model is “just right” in their prediction. This also gives a goal for

further development of models. Hopefully this will contribute to improvements in future models. We take the model results from published model archives without altering their physical input. We do not attempt to pinpoint which parametrization contributes to a particular model's incorrect climate gain. Nevertheless, knowing which model is more realistic can contribute to a better prediction of future warming from this subset of models.

- (7) *“The Earth’s climate response to the Sun’s output variations depends on the direct effects of changes in the total solar irradiance (TSI) and solar spectral irradiance (SSI) on atmospheric and surface heating rates, and on the indirect effects of SSI fluctuations and of variations in the fluxes of energetic particles on atmospheric composition. TSI, SSI, and energetic particles’ fluxes are found to vary at all time scales covered by direct measurements, and they are expected to vary also on longer time scales. For many years, climate models only accounted for TSI changes, but recent models have started to cover all the relevant solar irradiance and energetic particle contributions with increasing accuracy”.*

It could be that the Reviewer is not criticizing our calculation, but that we did not discuss these other solar forcing types. We did discuss them previously in the Methods. In the revised version we have now expanded that discussion.

Variations in the fluxes of energetic particles on atmospheric composition consist of corona mass ejection of energetic particles (mostly ionized hydrogen) and the effect of solar wind on cosmic rays. Most CMIP6 models are unable to account for these particle fluxes with the exception of a few, such as WACCM. Notable recent studies focusing on the impacts of galactic cosmic rays (GCRs) on climate include Calisto et al. [2011, Atmos. Chem. Phys., 11, 4547–4556, doi:10.5194/acp-11-4547-2011] and Rozanov et al. [2012, Surv Geophys, 33, 483–501, doi:10.1007/s10712-012-9192-0]. They showed using the SOCOL model that the GCR induced changes in the chemical species, such as NO_x, HO_x, O₃, are the biggest as measured in percentage (~10%) in the middle troposphere. However, since these chemical tracers are most abundant in the stratosphere and less abundant in the troposphere by at least an order of magnitude, the contribution of the tropospheric change to the total NO_x abundance is very small.

In our recent publication [Wang et al. 2020, Solar Phys, 295, 117, doi:10.1007/s11207-020-01685-1], we estimated the solar-cycle induced changes in NO₂ using ground-based observations, WACCM, and our own 1-D photochemical model. [Note that the variation of GCR is opposite to that of the solar cycle: when the solar cycle is stronger in the positive phase, GCR reaching the Earth’s surface is less because of the stronger shielding by the solar magnetic field; see our publication: Huang et al, 2022, Atmos. Environ., 268, 118824, doi:10.1016/j.atmosenv.2021.118824] The changes are largely in the stratosphere, with exception during rare nonlinear events such as tropopause folding in the mid-latitude. Thus, the overall solar-cycle (or GCR) induced

changes in the tropospheric NO_x is much less than other variabilities, e.g. anthropogenic pollution, and therefore we do not focus on its climate impact in the current study.

The aforementioned studies also suggested that the GCR-induced changes of stratospheric species could lead to a reduction in stratospheric O₃, which in turn could strengthen the polar night jets and subsequently impact the regional climate [e.g. in northern Europe]. Matthes et al. [2017, *Geosci. Model Dev.*, 10, 2247–2302, doi:10.5194/gmd-10-2247-2017] used one of the CMIP6 model, WACCM and obtained a weakened polar night jet in the solar maximum, consistent with those studies, potentially impacting regional climate. We are interested in globally averaged surface response.

There have been proposals of the cosmic rays affecting the troposphere via their forming condensation nuclei for clouds. In fact, it was suggested that the early twentieth century global warming was caused by the cosmic rays (e.g. Henrik Svensmark). The second author, Tung, was in a National Academy of Sciences team that examined this issue in 2013 and found no evidence supporting the proposal and evidence contradicting it. Later in 2017 a satellite called “CLOUD” was launched to study this issue. The results were that the cosmic rays are “unlikely to be comparable to the effect of large variations in natural primary aerosol emissions”.

We also studied the solar-cycle induced changes in electron density in a number of publications [Li et al., 2018, *J. Geophys. Res.*, 123, 848–861, doi:10.1002/2017JA024634], HO_x [Li et al., 2017, *Earth Space Sci.*, 4, 607–624, doi:10.1002/2017EA000283] and O₃ [Li et al, 2016, *Earth Space Sci.*, 3, 431–440, doi:10.1002/2016EA000199; Li and Tung, 2014, *J. Geophys. Res.*, 119, 5823–5835, doi:10.1002/2013JD021065].

In summary, these and the studies mentioned earlier suggest that while these particle fluxes may affect chemical composition in the upper atmosphere, their impact on surface global temperature response is likely small. Furthermore, even if the effect is not small, it does not affect our results as long as these variations follow the time variation of the TSI(t), though some may be anticorrelated. This is because we extract the total solar cycle response from all solar forcing and regressed against the TSI(t) for both models and observation. In observation, obviously the effect is included. In the few models which include particle flux forcing and SSI, the response should include it. Though if such particle forcing and or SSI is important, the denominator of the regression could be larger or smaller. But since both the models and observations used the same regression, the relationship between them does not change.

REVIEWERS' COMMENTS

Reviewer #2 (Remarks to the Author):

I do not have further comments for the authors. Their remarks on the response to my comments and the new version of the manuscript have answered all my comments and observations. Congratulations.

Reviewer #3 (Remarks to the Author):

I have reviewed the revised version of the manuscript "Solar Cycle as a Distinct Line of Evidence Constraining Earth's Transient Climate Response" by King-Fai Li and Ka-Kit Tung.

The authors have addressed the points raised on the previous version of their manuscript. They have also offered further explanations in their reply. I find the new version of the manuscript to complement the existing literature on the topic, and I recommend its publication.

I suggest authors considering the following revisions.

R102: "in a disk facing the sun at the top of the atmosphere." -> revise in order to introduce TSI based on its definition

R104: "trend in TSI" ->
revise into "long-term variability in TSI"

R106: "based on solar radio-flux at 10.7cm and other 10.7 proxies measured at Earth's surface" ->
revise into "based on proxies of solar dark and bright magnetic regions."

Citation to <https://ui.adsabs.harvard.edu/abs/2023arXiv230303046C/abstract> should be added.

R167 --41 -> unclear, revise the whole statement

R867: "Since the TSI varies by about 1 Wm^{-2} from solar min to solar max. The uncertainty is about 5%." -> unclear, revise the whole statement

R173: ERF not defined yet

R854: "TSI reconstruction is based mostly on 10.7cm flux measured at Earth's surface." -> in order to

more accurately report existing literature on the topic, revise into "Reconstruction of long-term variation of TSI is based on proxies of solar magnetic activity, mostly sunspot records and chromospheric indexes on centennial time scale."

REPLY TO REVIEWERS' COMMENTS

REVIEWERS' COMMENTS

Reviewer #2 (Remarks to the Author):

I do not have further comments for the authors. Their remarks on the response to my comments and the new version of the manuscript have answered all my comments and observations. Congratulations.

We very much appreciated the two reviews from Reviewer #2

Reviewer #3 (Remarks to the Author):

I have reviewed the revised version of the manuscript "Solar Cycle as a Distinct Line of Evidence Constraining Earth's Transient Climate Response" by King-Fai Li and Ka-Kit Tung.

The authors have addressed the points raised on the previous version of their manuscript. They have also offered further explanations in their reply. I find the new version of the manuscript to complement the existing literature on the topic, and I recommend its publication.

We appreciated the two reviews by Reviewer #3. We thank the Reviewer for the final positive recommendation.

I suggest authors considering the following revisions.

R102: "in a disk facing the sun at the top of the atmosphere." -> revise in order to introduce TSI based on its definition

*R104: "trend in TSI" ->
revise into "long-term variability in TSI"*

*R106: "based on solar radio-flux at 10.7cm and other
10.7 proxies measured at Earth's surface" ->
revise into "based on proxies of solar dark and bright magnetic regions."*

*Citation to <https://ui.adsabs.harvard.edu/abs/2023arXiv230303046C/abstract>
should be added.*

R167 --41 -> unclear, revise the whole statement

*R867: "Since the TSI varies by about 1 Wm⁻² from solar min to solar max. The uncertainty is about 5%." -> unclear,
revise the whole statement*

R173: ERF not defined yet

*R854: "TSI reconstruction is based mostly on 10.7cm flux measured at Earth's surface." -> in order to more
accurately report existing literature on the topic, revise into "Reconstruction of long-term variation of TSI is based on
proxies of solar magnetic activity, mostly sunspot records and chromospheric indexes on centennial time scale."*

We have followed all of Reviewer's suggestions for editorial changes.